# Subcortical-cortical dynamical states of the human brain and their breakdown in stroke

Chiara Favaretto ®[1,2] ✉, Michele Allegra[1,3,4], Gustavo Deco ®[5,6], Nicholas V. Metcalf[7], Joseph C. Griffis[7], Gordon L. Shulman[7,8], Andrea Brovelli ®[4] & Maurizio Corbetta ®[1,2,7,8,9] ✉

The mechanisms controlling dynamical patterns in spontaneous brain activity are poorly understood. Here, we provide evidence that cortical dynamics in the ultra-slow frequency range (<0.01–0.1 Hz) requires intact cortical-subcortical communication. Using functional magnetic resonance imaging (fMRI) at rest, we identify Dynamic Functional States (DFSs), transient but recurrent clusters of cortical and subcortical regions synchronizing at ultra-slow frequencies. We observe that shifts in cortical clusters are temporally coincident with shifts in subcortical clusters, with cortical regions flexibly synchronizing with either limbic regions (hippocampus/amygdala), or subcortical nuclei (thalamus/basal ganglia). Focal lesions induced by stroke, especially those damaging white matter connections between basal ganglia/thalamus and cortex, provoke anomalies in the fraction times, dwell times, and transitions between DFSs, causing a bias toward abnormal network integration. Dynamical anomalies observed 2 weeks after stroke recover in time and contribute to explaining neurological impairment and long-term outcome.

In the healthy brain, neuronal populations interact at multiple temporal scales, from hundreds of milliseconds to tens of seconds through interlocked rhythms[1–4]. While neuroscience has traditionally focused on fast neural spiking activity[5,6], more recent theoretical work and simultaneous recordings from thousands of neurons show that activity in the infra-slow frequency range (<0.1 Hz) recruits the majority of the brain's energy budget[7–9], and is behaviorally relevant[10–13]. In the human brain, infra-slow fluctuations can be easily measured with fMRI blood oxygenation level-dependent (BOLD) and EEG/MEG signals[14]. Infra-slow activity is organized in distinct spatiotemporal patterns known as resting state networks, formed by groups of regions showing temporally correlated activity (functional connectivity, FC) and co-activating during behavioral tasks[15,16]. More recently, it has been shown that this network structure reflects the long-time average ('static FC') of rapidly switching connectivity patterns ('dynamic FC' or dFC) which can be consistently observed with different analysis methods[17–19] and are significantly correlated with global behavioral traits (e.g. processing speed or fluid intelligence[1]). The mechanisms controlling the large-scale temporal coordination of infra-slow activity are unclear, particularly whether specific regions play a leading role in orchestrating global changes in connectivity patterns. A leading hypothesis is that shifts in brain states at rest, or during tasks, depend on highly interconnected cortical regions (hubs), e.g., precuneus, posterior cingulate cortex, and lateral prefrontal cortex, that flexibly interact at different

[1]Padova Neuroscience Center (PNC), University of Padova, via Orus 2/B, 35129 Padova, Italy. [2]Department of Neuroscience (DNS), University of Padova, via Giustiniani 2, 35128 Padova, Italy. [3]Department of Physics and Astronomy "Galileo Galilei", University of Padova, via Marzolo 8, 35131 Padova, Italy. [4]Institut de Neurosciences de la Timone UMR 7289, Aix Marseille Université, CNRS, 13005 Marseille, France. [5]Center for Brain and Cognition (CBC), Department of Information Technologies and Communications (DTIC), Pompeu Fabra University, Edifici Mercè Rodoreda, Carrer Trias i Fargas 25–27, 08005 Barcelona, Catalonia, Spain. [6]Institució Catalana de Recerca I Estudis Avançats (ICREA), Passeig Lluis Companys 23, 08010 Barcelona, Catalonia, Spain. [7]Department of Neurology, Washington University School of Medicine, 660S. Euclid Ave, St. Louis, MO 63110, USA. [8]Department of Radiology, Washington University School of Medicine, 660S. Euclid Ave, St. Louis, MO 63110, USA. [9]VIMM, Venetian Institute of Molecular Medicine (VIMM), Biomedical Foundation, via Orus 2, 35129 Padova, Italy. ✉e-mail: chiara.favaretto.2@unipd.it; maurizio.corbetta@unipd.it

points in time with different networks[20–22]. However, recent studies have also shown hubs in subcortical regions (basal ganglia[23], thalamus[23,24], hippocampus[20,25–27]). Whether cortical synchronization also relies on subcortical regions is still poorly known. Clinical work has shown FC dynamics alterations in a variety of non-focal conditions (neurodegeneration, consciousness abnormalities, schizophrenia, autism)[17,19,28–31], which suggests that even more pronounced alterations should occur in focal conditions. Focal lesions, such as those induced by stroke, provide an ideal testbed to study the relations between brain structure and dynamics, since they considerably amplify the natural range of inter-subject variability in anatomical as well as functional connectivity. Subcortical lesions produce widespread functional alterations of the 'static' network structure—an anomalous inter-hemispheric segregation and intra-hemispheric integration[32–34]. Recent studies indicate that lesions induce anomalies also at the dFC level, altering the dynamic balance between integration and segregation[35–37]. However, which structural changes determine these functional anomalies, and how the interplay between cortex and sub-cortex contributes to them, has never been thoroughly investigated.

In this work, we analyze dFC patterns ("dynamic functional states" or DFSs) in a large cohort of first-time stroke patients at different clinical stages. Our study has three aims. First, we wish to describe DFSs both at the cortical and subcortical level and determine whether cortical and subcortical state dynamics are linked. Second, we examine the relation between structural lesions and alterations of FC dynamics, in terms of the fraction times and dwell times of DFSs. We use machine learning methods to explain abnormalities of dynamic FC with lesion location and patterns of structural disconnection, either at the cortical or subcortical level[34,38,39] or in cortical-subcortical pathways[27,40,41]. Finally, guided by previous results on the behavioral relevance of lesion location[38,42], structural disconnections[34,42], and static FC abnormalities[33,43], we test whether information about FC dynamics enhances explanation of behavioral deficits and acute-to-chronic explanation of behavioral recovery.

## Results
### Definition of dynamic functional states
Control and stroke subjects were identified from the Washington University Stroke cohort (https://cnda.wustl.edu/app/template/Login). To obtain reliable dFC estimates at the individual level, we analyzed only subjects with at least 300 TR (600 s) of valid signals after pre-processing and censoring. This criterion identified 20 controls with two scan sessions 3 months apart, and 47 patients with first-time strokes with scans at three time points (2 weeks, 3 months, 12 months). The lesions frequency map of the stroke group shows that most lesions involve the deep middle cerebral artery distribution with damage of the basal ganglia and subcortical white matter (SI-Fig. 1). Fewer than 20% were cortical lesions. This distribution matches previous pro-spective cohorts of stroke lesions[38,44,45] (SI-Tables 1–3).

We analyzed dFC through the most straightforward approach: the sliding-window temporal correlation (window width = 60 s, window step = 2 s) followed by eigenvector decomposition and clustering (see "Methods" section and Fig. 1 for flowchart) to define a set of con-nectivity states that continuously activate and deactivate over time (DFSs), as in refs. 46–48. All projected data from controls (CTRs) and patients (PATs) (at all time points) were concatenated and clustered in time (with K-means algorithm), yielding a limited number of DFSs. We ran K-means between K = 2 and K = 10. The optimal solution was K = 5 selected through Silhouette and Davies-Bouldin indexes (SI-Fig. 2). We performed several control analyses to ensure their robustness both in terms of the size of the temporal window used for the calculation, and the number of states (K = 2 ÷ 10). We also showed that the DFSs were representative, in each window, of the dynamic FC from which they were derived (see SI paragraph S1–S4, S6 and SI-Figs. 3–7, SI-Fig. 11).

## Dynamic functional states (DFSs) capture cortical and sub-cortical interactions
Five DFSs described the dynamic functional connectivity changes in healthy controls and stroke patients. We used several representations to illustrate these functional states in cortical and subcortical regions.

Figure 2 visualizes DFSs in matrix form (Fig. 2a) and through a circular graph representation (Fig. 2b). Positive weights indicate positive co-modulation, whereas negative weights indicate negative co-modulation between brain regions. We characterized DFSs in terms of the most common static FC biomarkers observed in stroke[32,34,43], namely: (1) the average homotopic inter-hemispheric connectivity (2) the average intra-hemispheric connectivity between task-positive (DAN) and task-negative (DMN) regions, as a measure of network integration (3) the overall Newman's modularity among cortical net-works, as a measure of segregation (Fig. 2c).

Figure 3 focuses on cortico-subcortical interactions, which are illustrated either with subcortical regions vs. cortical networks in matrix form (Fig. 3a), or as brain surface/volume maps (for cortical/subcortical regions respectively) plotting the first eigenvector of each DFS (Fig. 3b). To facilitate analysis of cortico-subcortical interactions, we performed principal component analysis (PCA) on the leading eigenvector of subcortical connectivity, identifying two main sub-cortical components (SCs): SC1 loads on cerebellum and subcortical nuclei: thalamus, caudate, putamen, nucleus accumbens, and globus pallidus. SC2 loads on 'limbic' regions like amygdala and hippocampus. By construction, these two components are not correlated. Moreover, they always correlate in opposite directions with different cortical networks.

Each DFS is characterized by a different set of cortical and cortical-subcortical interactions. Herein, we provide a description of each state based on these criteria. DFS1 is very similar to the healthy static FC, with high homotopic connectivity ($\rho_z = 0.45 \pm 0.0013$, where $\rho_z$ indi-cates correlation coefficient after z-Fisher transformation), large negative DAN-DMN connectivity ($\rho_z = -0.25 \pm 0.002$), and high modularity ($0.22 \pm 0.0006$). Sensory-motor-attention networks (visual: VIS, sensorimotor: SMN, auditory: AUD; control: CON, dorsal attention: DAN) are positively correlated, and negatively correlated with the default mode network (DMN) (Fig. 2a). These patterns cor-respond to the well-known separation between task-negative and task-positive networks[49]. In-between stand high-level cognitive networks, such as the ventral attention (VAN) and fronto parietal network (FPN) that are weakly correlated with either task-positive or task-negative networks, and are expected to exhibit more flexible interactions and more individual variability[50,51]. When we consider cortical-subcortical interactions, DFS1 is characterized by a positive correlation between DMN and limbic nuclei, SC2 (Fig. 3a, b), which in turn are negatively correlated with sensory-motor-attention networks.

DFS2 is very similar to the 'pathological' static FC observed in stroke, with low homotopic connectivity ($\rho_z = 0.37 \pm 0.002$), nearly zero DAN-DMN connectivity ($\rho_z = -0.07 \pm 0.002$), and low mod-ularity ($0.21 \pm 0.0007$). This state is characterized by a strong inte-gration of cognitive networks (DAN, VAN, CON, FPN) and a strong negative coupling of the VIS network with other networks. AUD, SMN, and DMN maintain strong internal correlation, but remain relatively independent.

DFS3 is characterized by a high homotopic connectivity ($\rho_z = 0.47 \pm 0.0014$) and a high modularity ($0.24 \pm 0.0006$), similar to DFS1. However, it does not show a strong (negative) correlation between DAN and DMN ($\rho_z = -0.09 \pm 0.002$). Instead, it captures a negative correlation between a sensory-motor cluster (VIS, SMN, AUD) and a cognitive cluster (FPN, DMN, DAN, VAN). Like DFS1, DFS3 is characterized cortically by the well-known segregation between sensory-motor networks (VIS-AUD-SMN) and DMN. However, cortico-subcortical interactions are very different in the two states: the cou-pling pattern between SC1–SC2 and sensorimotor/DMN is opposite.

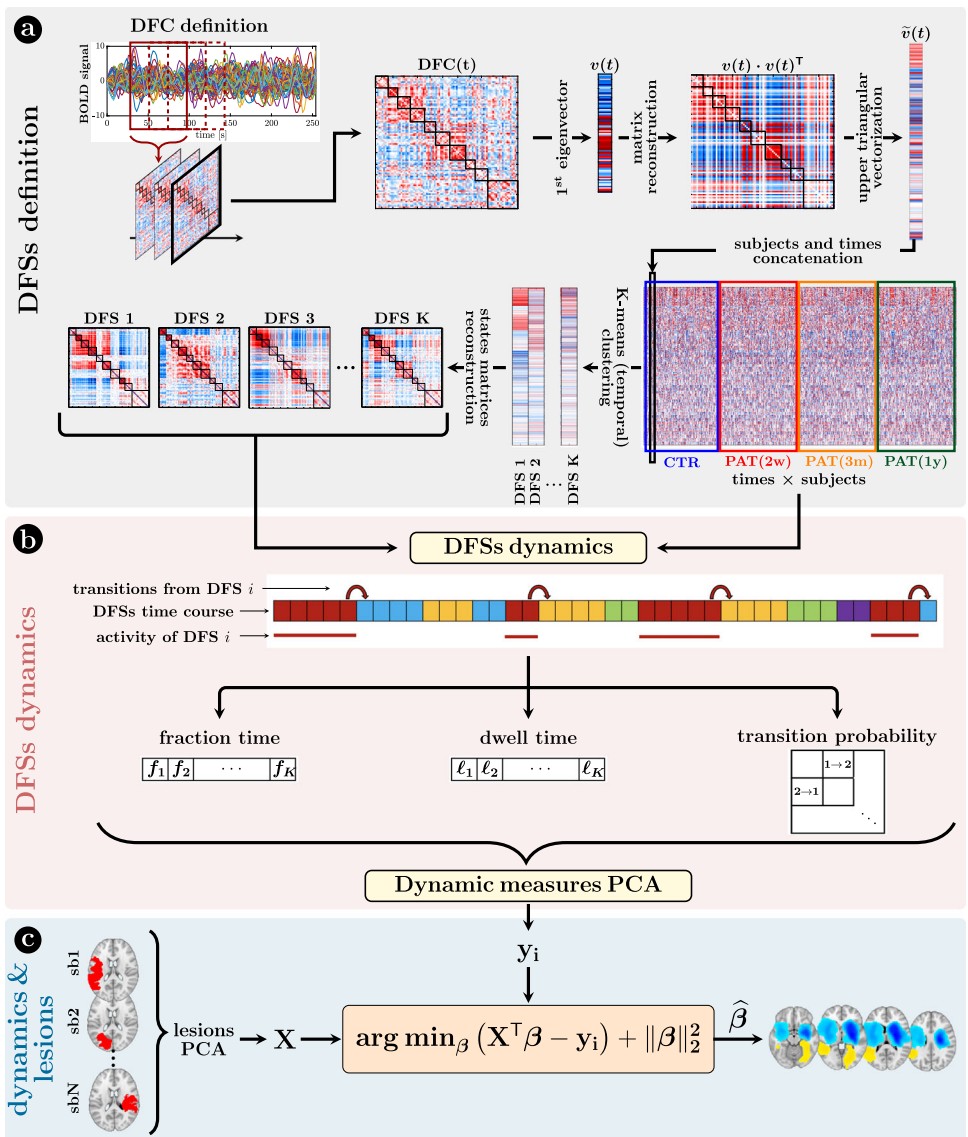

**Fig. 1 | Methods (dynamical functional states). a** Definition of the dynamic functional states (DFSs): (i) at first, the time course of each subject was divided into 270 time-windows of width = 30 TR (600 s) and step = 1 TR. The z-Fisher transform of the Pearson's correlation coefficient among regions was computed at each sliding window, to estimate the Dynamical Functional Connectivity (DFC). Then, (ii) each DFC matrix was approximated by projecting on the leading eigenspace defined by the first eigenvector $v_i$. As the eigenvectors are defined less than the sign, we avoided this issue by reconstructing the square matrix $v_i \times v_i^T$. After that, (iii) the upper triangular part of these rank-one DFC matrices was vectorized and concatenated across windows and subjects, in order to finally apply a time-wise K-means clustering algorithm with correlation distance and 20 replicates (iv) to define a set of K spatial DFSs. Silhouette and Davies-Bouldin algorithms were used to search for the optimal number of DFSs. Several choices

of K (from 2 to 10) were used for supplementary analyses and comparisons. **b** The K-means clustering associated each sliding window to a specific DFS, thus for each subject we obtained a discrete time series $x(n)$, with $n = 1,...,270$, where each discrete value (between 1 and K) indicated the active state at that time point. From these time courses it was possible to evaluate three different dynamical measures for each state, namely the fraction time, the dwell time, and the transition probability. To analyze the relationship among dynamical measures in healthy condition, we performed a Principal Component Analysis (PCA) over all the dynamical measures. **c** The projection of the sub-acute patients' dynamical measures onto the PCs space, and the anatomical brains lesions were used as input for a Ridge Regression algorithm, aimed at identifying the lesion's location that better characterized specific dynamic impairments. A similar approach has been used with structural disconnections.

SC2 (limbic) correlates positively with the DMN in DFS1, but negatively in DFS3. Correspondingly, SC2 shows no correlation with sensory-motor networks in DFS1, but a positive correlation in DFS3. In contrast, SC1 (nuclei) shows no correlation with DMN in DFS1, but a positive correlation with DMN and a negative correlation with sensory-motor networks in DFS3 (Fig. 3c, d).

DFS4 shows intermediate values of homotopic connectivity ($\rho_z = 0.42 \pm 0.0014$), DAN-DMN connectivity ($\rho_z = -0.14 \pm 0.002$), and modularity ($0.22 \pm 0.0006$). In this state, we observe anti-correlation between a VIS-DAN-FPN cluster and a SMN-AUD-CON-VAN-DMN cluster. Interestingly, in DFS4 all subcortical regions

show positive correlation and appear strongly uncorrelated from cortex.

Finally, DFS5 shows intermediate values of homotopic ($\rho_z = 0.42 \pm 0.0015$) and DAN-DMN ($\rho_z = -0.09 \pm 0.002$) connectivity, and a very low value of modularity ($0.20 \pm 0.0007$). Indeed, it reflects another state of integration among almost all networks (like DFS2), except VIS and DMN that remain more segregated. DFS5 differs from DFS2 for the absence of the negative correlation between VIS and all other networks.

In summary, we identified a set of spatial maps of inter-regional correlation alternating over time (DFSs) characterized by different

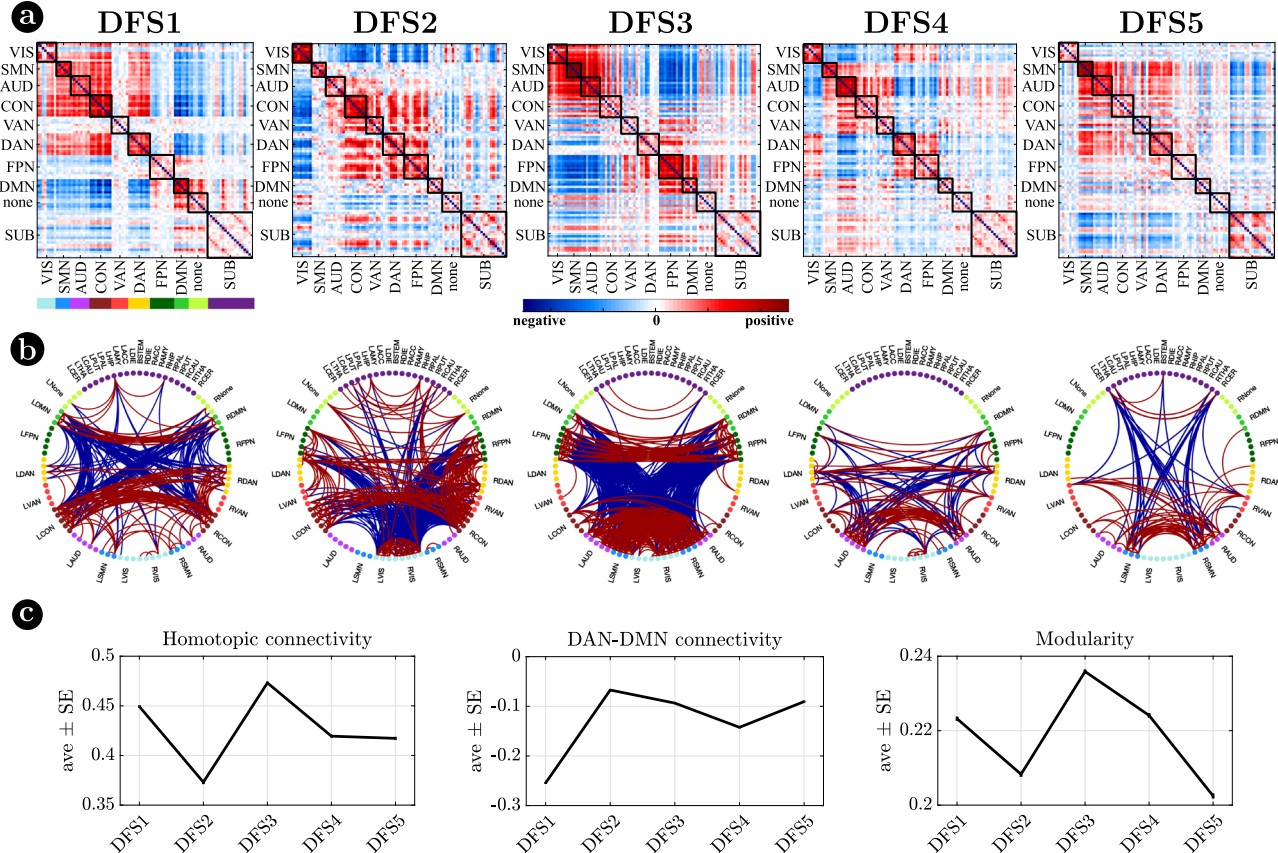

**Fig. 2 | Dynamic functional states (DFSs). a** Representation of the 5 DFSs in matrix form (positive and negative weights are red and blue, respectively). **b** The same DFSs are described through a circular graph representation. In each column, the positive (red) and negative (blue) strongest links for each state are represented. Network belonging is color-coded. **c** Average and standard error of (left) the average homotopic inter-hemispheric connectivity within each network; (center) the average connectivity between dorsal attention network (DAN) and the default mode network (DMN) regions, as a measure of task-positive and task-negative network integration); (right) the overall Newman's modularity among cortical networks. Data are reported as mean values +/− SEM. Source data are provided as a Source data file.

cortical and subcortical connectivity patterns. An important insight is that, in each DFS, different cortical clusters connect with a specific set of subcortical nuclei (Fig. 3a, b). In fact, changes in cortical states were temporally coincident with shifts in subcortical connectivity. Figure 4a shows exemplar time courses of the leading eigenvector of cortical connectivity, projected onto different networks (top), and the two principal components of the leading eigenvector of subcortical connectivity (bottom), during switch between DFSs. Coincident cortico-subcortical reorganization is plainly appreciable. Importantly, the reorganization of subcortical and cortical patterns was generally synchronized irrespective of the definition of a DFS switch. Indeed, we evaluated connectivity shifts, defined as connectivity differences between pairs of consecutive sliding windows, separately for cortical and subcortical regions. We found that subcortical shifts were positively correlated with network cortical shifts (all correlations >0.5). Both cortical and subcortical connectivity shifts showed (in absolute value) a heavy tail distribution, with more frequent low differences (Fig. 4b left). Thus, we defined a jump when a large connectivity difference occurred (0.29, corresponding to the top 5% values). Then, we tested the simultaneity of cortical and subcortical reorganization by comparing the probability that cortical and subcortical jumps occur simultaneously (estimated as $P$(subcortical changes|cortical changes)) under the null hypothesis of independent processes, and in the observed data (see "Methods" for details on this analysis). We evaluated these for each subject, and we compared the two distributions through the Wilcoxon rank test. For all networks, we found that the observed conditioned probability was significantly

larger than the probability under the null hypothesis (all $p < 10^{-40}$, Bonferroni corrected for 9 networks), supporting the idea of synchronous cortical and subcortical shifts (Fig. 4b right). It is important to highlight that in this analysis we considered all sliding windows, not just those defining DFS boundaries. Therefore, the time course synchronization analysis was independent of the DFS definition.

Importantly, the observed coordination between cortical and subcortical dynamics does not depend on the specific subcortical parcellation used. We replicated our original analyses (based on the Freesurfer parcellation) with a more recent subcortical parcellation[52]. Tian et al.[52] developed four subcortical parcellations with increasing levels of resolution (16, 32, 50, or 54 regions, respectively). We limited our analysis to the lowest (16 regions) and highest (54 regions) resolution parcellations. Detailed results are presented in the Supplementary Information (SI paragraph S7 and SI-Figs. 12, 13). The choice of parcellation did not influence our three main findings: (1) the 'antagonistic' dynamics of basal ganglia vs limbic regions, represented by two anticorrelated principal components of subcortical dynamic FC; (2) the observation that different DFS are associated with different patterns of cortical/subcortical interactions, as shown by different patterns of connectivity between the main subcortical clusters and cortical networks; and (3) the coordination between cortical and subcortical dynamics, as shown by simultaneous cortical/subcortical FC shifts. Qualitatively, the main difference between results in the two parcellations is related to the thalamus. While in the Freesurfer parcellation, used in the original analysis, the thalamus essentially grouped with the basal ganglia, the new parcellation yields a more

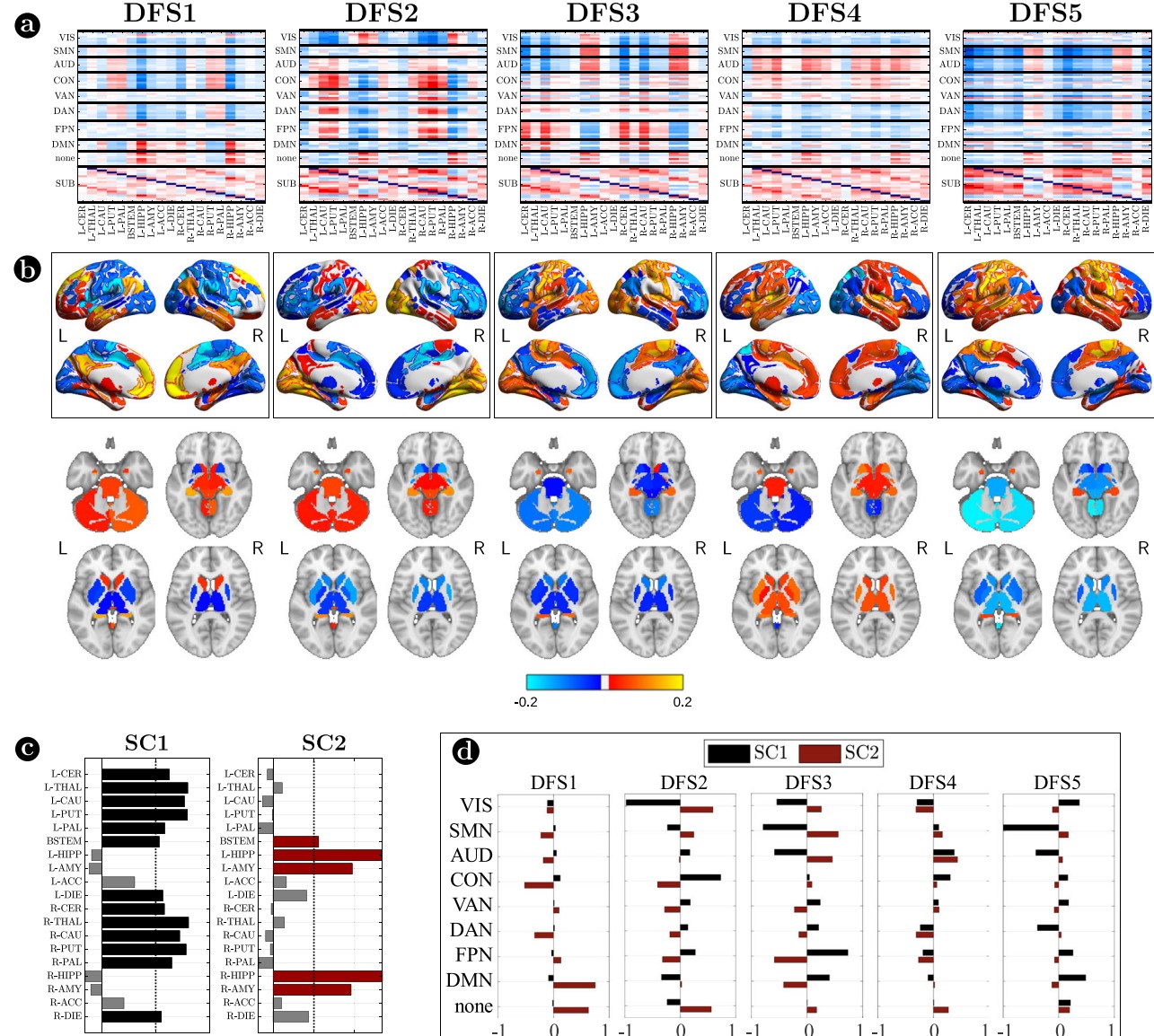

**Fig. 3 | Cortico-subcortical interaction in the dynamic functional states (DFSs). a** Matrix representation of cortico-subcortical interaction. This is a zoom of Fig. 2a. Positive and negative values are indicated with red and blue, respectively. **b** Surface and volume projection of the first eigenvector of each DFSs. Cortical regions are shown in surface (top), while subcortical regions in volume (bottom). **c** Loading of each subcortical regions in the two main subcortical components. A threshold of 0.2 has been used. **d** Average connectivity between each subcortical component (SC1 and SC2) with each cortical network in the different DFSs. Source data are provided as a Source data file.

nuanced picture, hinting at a functional split between different parts of the thalamus: the anterior portion of the thalamus groups with the basal ganglia, whereas the posterior portion cannot be clearly affiliated to either of the two clusters (basal ganglia/limbic). A fine-grained analysis of the relation between thalamic nuclei and DFS is left for future work.

### Sub-acute stroke causes a DFS imbalance with a bias toward integration that recovers over time

**DFSs imbalance in stroke patients.** Next, we employed dynamical measures related to the alternation of DFSs to study how stroke lesions affect these dynamic features ("Methods", Fig. 1). By construction, only one DFS can be active in each sliding window. Therefore, the dynamic of the functional connections can be described in terms of a single time series of discrete values (from 1 to 5), each associated to a DFS. Three measures were extracted to characterize the dynamic of DFSs, namely the fraction time ($f$), the dwell time ($\ell$) and the transition

probability (e.g., transition from DFS1 to DFS2: DFS1>2), which describe the number of times each state is active, the duration of each state and the probability to switch from one state to another ("Methods" and Fig. 1 for details), respectively.

The characterization of DFSs in terms of the most common static FC stroke biomarkers suggest that DFSs alterations may be more sensitively detected in patients with more severe static FC impairment. Accordingly, we divided stroke patients with severe or mild static (average) FC impairment at 2 weeks and performed all dynamic analyses with three groups: healthy controls, stroke patients with severe or mild FC impairment at 2 weeks. To that effect, we performed a spatial PCA to find a component summarizing the static FC abnormalities explaining the largest portion of variance over patients. To avoid biases in patient selection, this PCA was run in an independent sample of 67 sub-acute patients not suitable for the dynamic analysis. The weights of this summary component (ST) identified two groups of patients: with more severe (ST > 0) ($n = 18$) or milder (ST < 0) ($n = 29$) static FC changes

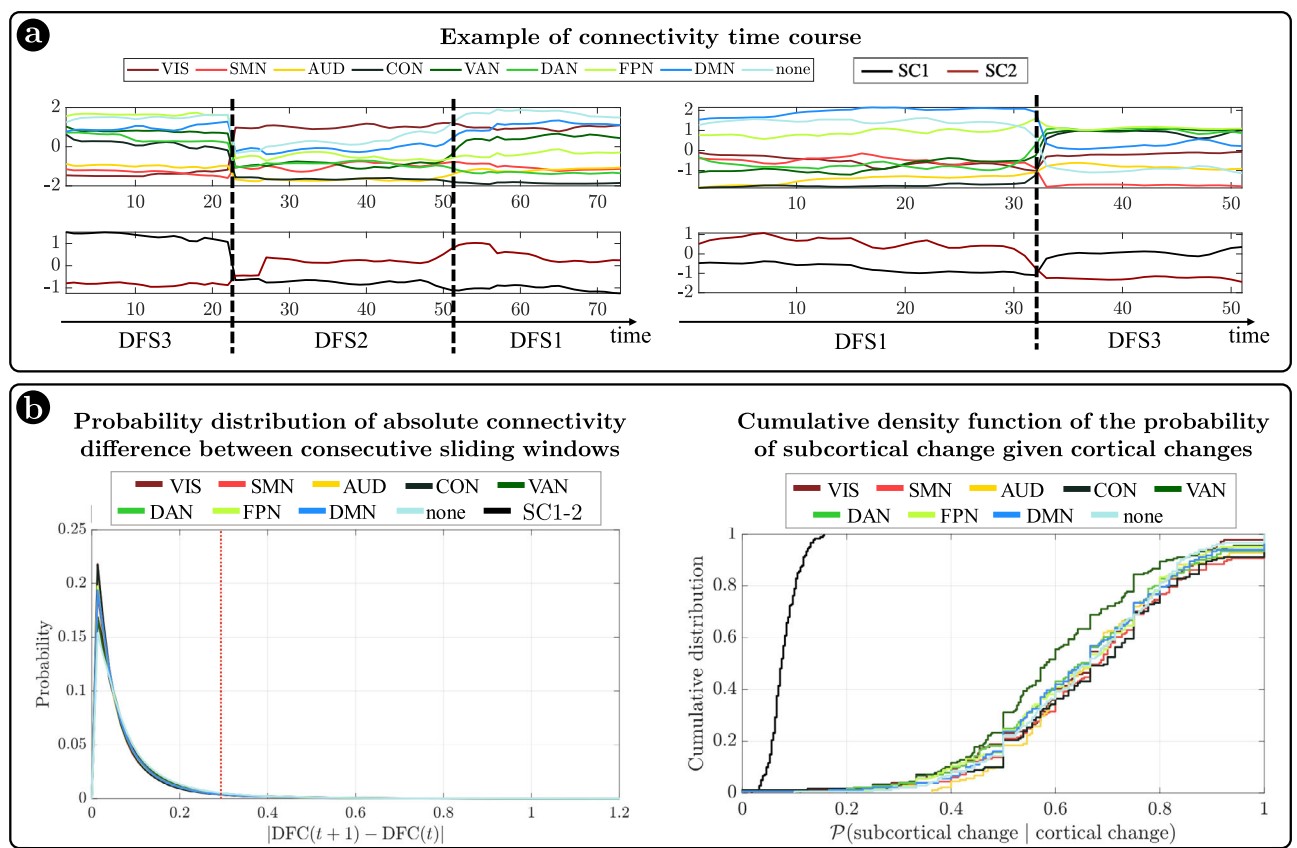

**Fig. 4 | Cortical and subcortical dynamics. a** Two examples of average connectivity during time for cortical networks (top) and subcortical clusters (bottom). The vertical dashed lines indicate the switching between dynamic functional states (DFSs). **b** (Left) Probability distribution of the absolute values of connectivity differences between consecutive sliding windows. Each line represents a different network. Right) Cumulative density function of the conditioned probability of subcortical connectivity reorganization, given a cortical connectivity reorganization. Each colored line relates to a different cortical network. The black line shows the cumulative density function under the null hypothesis of independence between cortical and subcortical changes. Source data are provided as a Source data file.

(for details SI paragraph S8 and SI-Fig. 15). In what follows, we will refer to these two groups as 'severe' and 'mild' patients, as patients with ST > 0 were overall more severe in their neurological impairment than patients with ST < 0 (mean NIHSS score: 7.23 (ST > 0) vs 2.37 (ST < 0); t test: $T = 4.02$, $p = 2.6 \times 10^{-4}$) as shown in SI paragraph S8. The main dynamical difference between control subjects and sub-acute patients was the fraction time ($f$) of the different DFSs (Fig. 5a top graphs). The control population showed a uniform distribution of DFSs' fraction time, with the exception of DFS5, which was significantly less frequent than all the other states except DFS4, as assessed through a nonparametric permutation test (mean ± standard error: $f_1 = 0.25 \pm 0.03$; $f_2 = 0.22 \pm 0.03$; $f_3 = 0.23 \pm 0.02$; $f_4 = 0.18 \pm 0.02$; $f_5 = 0.11 \pm 0.01$; $f_5 < f_1$ ($t = -4.25$), $f_5 < f_2$ ($t = -3.50$), $f_5 < f_3$ ($t = -4.50$): all $p = 10^{-3}$, Bonferroni corrected). Similarly, the dwell time ($\ell$) was similar across all DFS except for a significantly longer duration of DFS1 as compared to DFS5 ($\ell_1 > \ell_5$, $t = 3.23, p = 0.02$, Bonferroni corrected) (mean ± standard error: $\ell_1 = 13.97 \pm 1.44 TR$; $\ell_2 = 12.16 \pm 1.46 TR$; $\ell_3 = 12.59 \pm 1.13 TR$; $\ell_4 = 12.47 \pm 1.24 TR$; $\ell_5 = 8.40 \pm 0.89 TR$).

We tested the effect of DFSs and groups (controls, sub-acute severe patients, and sub-acute mild patients) in the fraction times through a generalized linear mixed effect model (GLME) with Poisson distribution, with DFS and group as factors. We found a significant interaction effect ($F = 211.13, p = 0$) and both single main effects of DFSs ($F = 151.95, p = 0$) and groups ($F = 168.29, p = 0$). Note that this result may be affected by collinearity, as fraction times for different DFS are not independent. However, we conducted post hoc analyses for each DFS separately (with non-parametric permutation tests), finding two different patterns of abnormalities for severe and mild patients. Severe

patients manifested an abnormal increase of DFS2 ($f_2 = 0.36 \pm 0.06$; $t = 2.30, p = 0.02$, FDR corrected for 15 comparisons) and marginally DFS4 ($f_4 = 0.28 \pm 0.05$; $t = 1.84, p = 0.051$, FDR corrected), at the expense of DFS1 ($f_1 = 0.13 \pm 0.02$) and DFS3 ($f_3 = 0.11 \pm 0.02$) that were significantly less frequent than in CTRs (DFS1: $t = -3.50, p = 0.003$ FDR corrected; DFS3: $t = -4.53, p = 0.0003$, FDR corrected). Therefore, severe patients had an anomalous under-expression of segregated states 1 and 3 and overexpression of integrated state 2. In contrast, mild patients showed an imbalance between the two integrated states, with an abnormal increase of DFS5 ($f_5 = 0.20 \pm 0.04$; $t = 2.20, p = 0.024$, FDR corrected for 15 comparisons) occurrences, and an anomalous decrease in DFS2 ($f_2 = 0.15 \pm 0.02$; $t = 1.98, p = 0.046$, FDR corrected). In general, measures of dwell time (Fig. 5a bottom graphs) replicated the patterns observed with fraction time but were less sensitive.

Fraction time and dwell time measures do not consider dynamical switches among DFSs that are defined by the transition probability (Fig. 5b). We found a significant interaction ($F = 2.99, p < 10^{-8}$) and a main effect of transitions ($F = 1.83, p = 0.015$) through a GLME with Normal distribution. As compared to controls, more severe sub-acute patients were characterized by more frequent bidirectional transitions between DFS4 and DFS2 (DFS2>4: $t = 3.16, p = 0.004$; DFS4>2: $t = 3.20, p = 0.004$, FDR corrected for 20 comparisons), and fewer transitions from DFS4 to DFS3 (DFS4>3: $t = -3.27, p = 0.004$, FDR corrected). Moreover, bidirectional anomalies in transitions were found between DFS1 and DFS5 for mild patients (increased DFS1>5: $t = 2.25, p = 0.05$ and DFS5>1: $t = 2.50, p = 0.03$, FDR corrected). In summary, the transition analysis showed that more/less frequent states

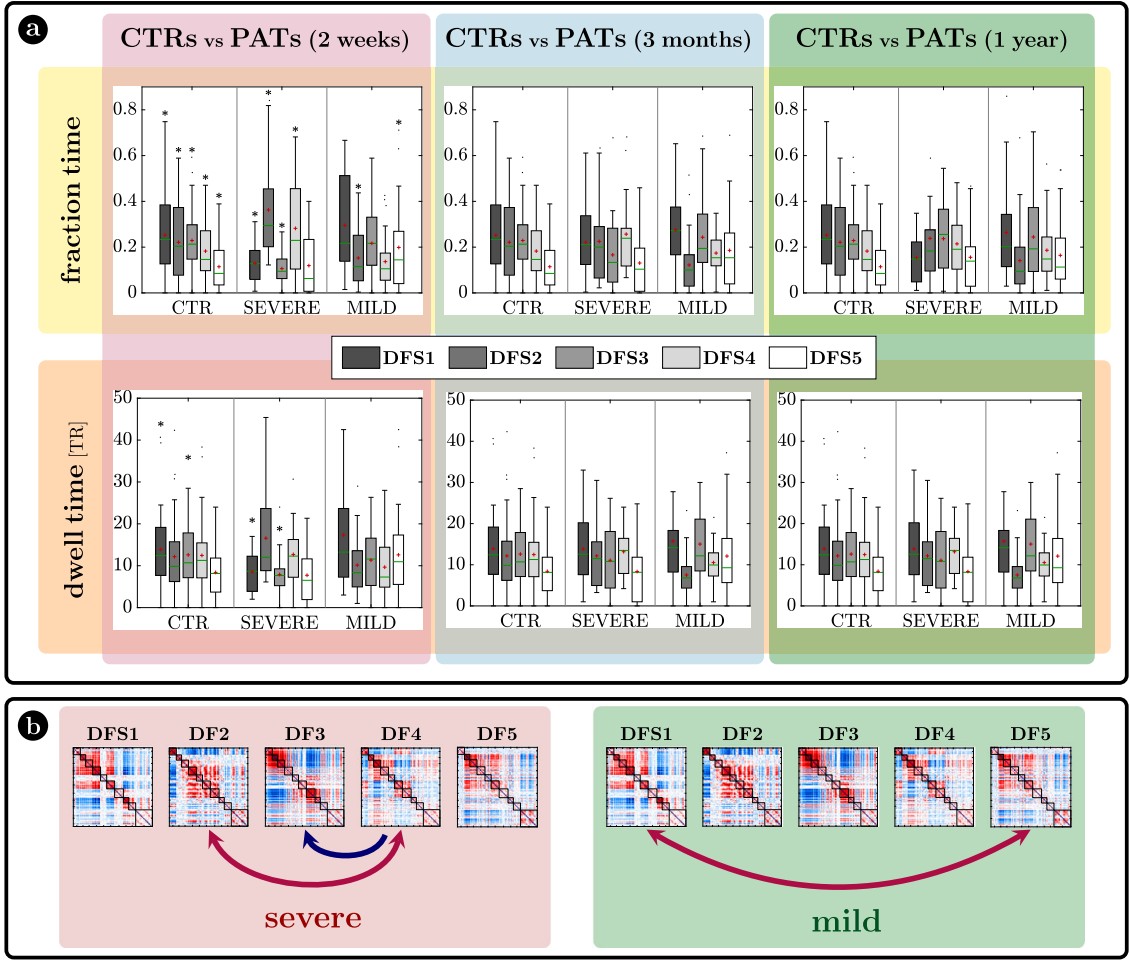

**Fig. 5 | Results. a** Fraction time and lifespan. This figure represents two dynamical measures related to the dynamic functional states (DFSs), namely the fraction time (top) and the average dwell time (bottom). Comparisons between healthy controls (CTRs) and stroke patients (PATs) at different conditions are represented. To be noticed: (i) fraction times and dwell times show similar patterns, but fraction times are more sensitive to identify group differences, and (ii) all dynamic impairments identified at the sub-acute stage, recovered in the chronic stage. The significance between each pair of groups has been tested independently for each of the 5 DFSs through one-sided non-parametric permutation tests, and false discovery rate (FDR) correction for 15 comparisons. The symbol * indicates *p*-value < 0.05 after FDR correction. For each panel: *n* = 40 (CTRs), 18 (PATs severe), 29 (PATs mild). On each box: the central green line indicates the median, the red cross indicates the mean, and the bottom and top edges of the box indicate the 25th and 75th percentiles, respectively. The whiskers extend to the most extreme data points, not considered outliers (plotted individually using a dot). **b** Graphical representation of the significant differences in transition probabilities between CTRs and patients at the sub-acute stage: red arrows represent increase in probability, while blue arrows stand for decreased transition probabilities. Source data are provided as a Source data file.

in stroke are also visited more/less frequently, e.g., DFS2 for severe patients and DFS5 for mild patients. Notably, all dynamic abnormalities in fraction time, dwell time, and transition probability observed at the sub-acute stage recovered over time at 3 and 12 months (Fig. 5). In summary, more severe stroke patients are characterized by more frequent network integration states (DFS2,4), and more transitions towards them, with a corresponding decrease of network segregation states (DFS1,3). In addition, there is a clear difference between stroke patients with severe or mild FC anomalies, with the latter group preferring to spend more time in DFS5 than DFS2. Importantly, sub-acute alterations in state dynamics recovered at 3- and 12-months post-stroke.

In a control analysis (SI paragraph S5, SI-Fig. 8), we verified that these results are not a consequence of motion scrubbing. We considered both a more stringent and a more liberal scrubbing threshold (corresponding respectively to a higher and a lower number of censored frames) and we observed no qualitative impact on the results.

### Subcortical lesions and cortico-subcortical disconnection explain abnormal FC connectivity dynamics

Having established that dynamic connectivity abnormalities occur in stroke patients acutely and recover over time, we were interested in

establishing the anatomical basis of these alterations both in terms of lesion location and structural disconnection. We employed a machine learning strategy (ridge regression) to explain the degree of dynamic FC impairment from lesion location and the related structural disconnection of white matter pathways ("Methods", Fig. 1).

Given that individual dynamic measures are correlated, we used PCA to summarize the main dynamical feature abnormalities. We performed a PCA on fraction times, dwell times, and transition probabilities (30 dynamical features per subject). Three dynamic PCs (Dyn-PC) explain about 43% of the variability across subjects. We focus on these components since each of the subsequent principal components (from the fourth onwards) explains only a small fraction (<5%) of the total variance. Specifically, Dyn-PC1 loads positively on the dynamic measures related to DFS1, and negatively on those related to DFS2 and DFS5. Dyn-PC2 loads positively on DFS3, and negatively on transitions related to DFS2 and DFS4. Finally, Dyn-PC3 loads mostly on DFS4 and transitions between DFS4 and DFS5 (see SI paragraph S9 and SI-Fig. 16).

The ridge regression models the individual contribution of lesion location/volume or structural disconnection to the pattern of dynamic impairment captured by the Dyn-PC scores ("Methods", Figs. 1 and 6).

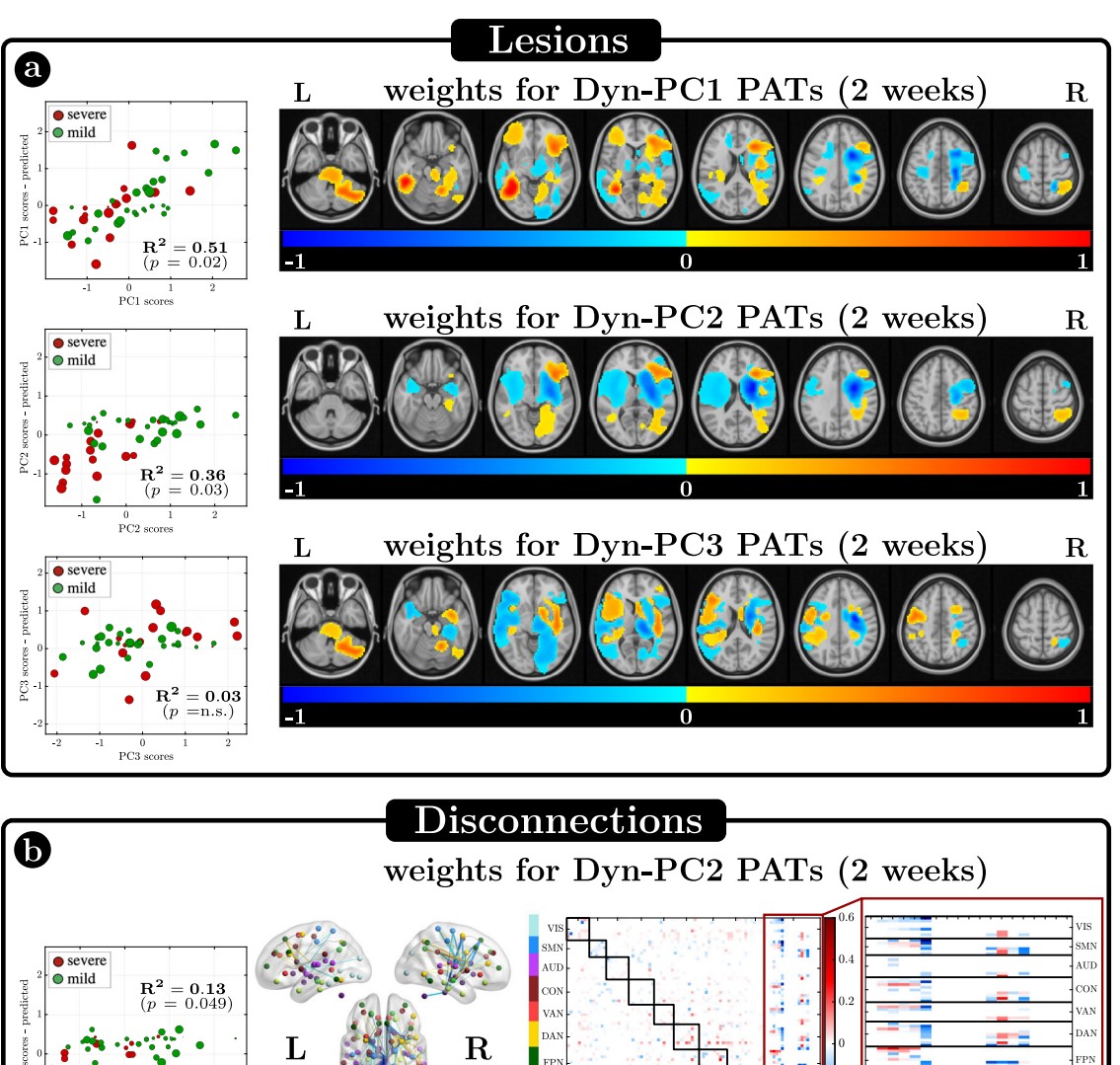

**Fig. 6 | Relationship between dynamic principal components (Dyn-PCs) and lesions or disconnections. a** Results of the Ridge Regression (RR) algorithm aimed at identifying possible existing relationships between the scores of the dynamical PCs and the anatomical lesions. On the left, the scatter plots between real and estimated values are shown, for the 3 Dyn-PCs. Each dot is a patient, whose stroke severity (severe or mild) is color-coded and whose lesion size is described by the dot dimension. $R^2$ is the amount of variance explained by each model, and $p$ the model significance. Only Dyn-PC1 and Dyn-PC2 are significantly described by RR models. On the right, the estimated optimal weights are represented, after normalizing w.r.t. their maximum absolute value. **b** Results of the Ridge Regression (RR) algorithm aimed at identifying possible existing relationships between Dyn-PC2 and the structural disconnections. On the left, the scatter plot between real and estimated values is shown. On the right, the significant disconnection weights are represented both in matrix form (right) and projected into the brain (left). Source data are provided as a Source data file.

Specifically, we used (i) the lesion map or (ii) the disconnection matrix of each subject as a regressor to estimate each dynamical PC separately. Like a common regression model, the ridge regression aims to assign a weight to each voxel (or link), indicating its contribution in explaining a specific Dyn-PC value. For instance, a positive weight indicates that a lesion in that voxel (or a disconnection of that structural link) is statistically associated with a positive value in that dynamical PC, and vice versa (see "Methods" for implementation and details).

The topography of lesions is significantly related to the variability of dynamic FC features (Fig. 6a).

Dyn-PC1 scores are high (positive, more segregation) in patients with cortical lesions but low (negative, less segregation) in patients with white matter lesions—typically severe patients, who tended to have subcortical lesions. Voxels associated with large values of Dyn-PC1 are hence grouped in several separated clusters, mirroring the heterogeneity of cortical lesions ($R^2 = 0.51$, $p = 0.02$ for Dyn-PC1). Dyn-PC2 scores are low (negative, less segregation) for lesions in the subcortical white matter and basal ganglia ($R^2 = 0.36$, $p = 0.03$ for Dyn-PC2). No significant models explain Dyn-PC3 scores.

The analysis of structural disconnection shows a significant relationship with Dyn-PC2 ($R^2 = 0.13$, $p = 0.049$) (Fig. 6b). The

# Likelihood ratio test results between Static and Dynamic PCs

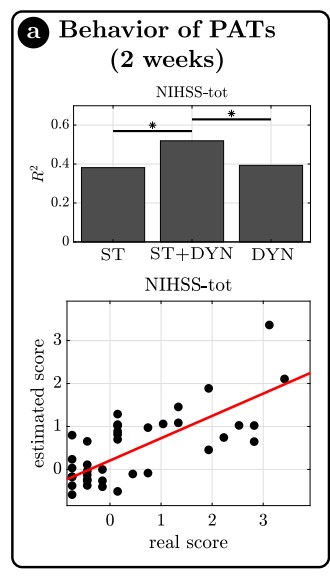

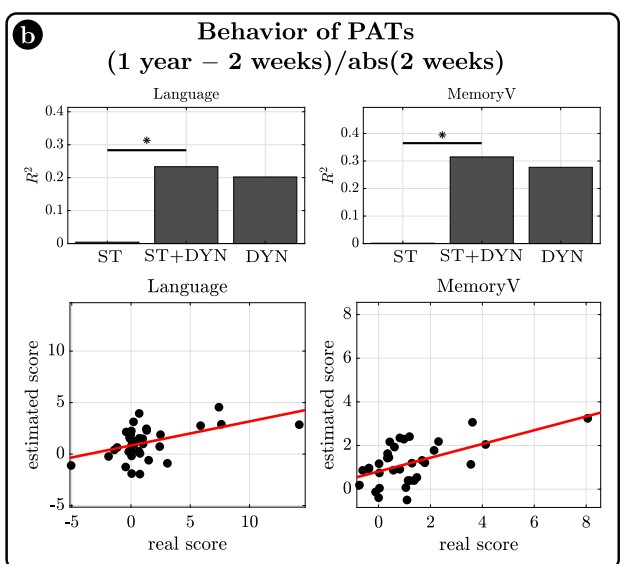

**Fig. 7 | Relationship between Dynamic measures and behavior.** The ratio likelihood test was used to test whether the addition of dynamic information to the static measures would significantly increase the ability of a generalized linear model to describe behavioral deficits (in terms of explained variance). Static measures were represented by static principal component (ST) values, while dynamic information was represented by dynamic principal components. **a** At first behavioral scores of sub-acute patients were considered. When dynamical PCs were used as dynamic regressors, only the global measure of behavioral impairment (NIHSS-total) resulted to be better estimated by the combination of static and dynamic regressors, than only static ones. The bar plot shows the $R^2$ for the reduced (only static, ST) model, for the unreduced (static and dynamic, ST+DYN) model, and for the reduced model (only dynamic, DYN). **b** As a second step, we used static and dynamic measures at sub-acute stage, to explain the difference in behavioral scores from 2 weeks to 1 year. When dynamical PCs were used as dynamic regressors, the explanation of score changes in Language and Visual Memory task was significantly better. Source data are provided as a Source data file.

corresponding scatter plots, maps, and disconnection matrix show that structural disconnection between brainstem/pallidum/putamen/thalamus and multiple cortical regions (DMN, DAN, CON, SMN) are associated with low Dyn-PC2 values, hence stronger network integration.

Notably, neither damage of the polymodal association cortex nor long-range association cortico-cortical pathways are significantly related to abnormalities in cortical dynamics or transitions among cortical states. While this negative result is preliminary given the low number of lesions at each cortical location, our findings support the importance of subcortical activity and basal ganglia/thalamo-cortical communication in controlling cortical dynamics in the 0.1 Hz temporal scale. This is also consistent with the temporal synchronization of cortical-subcortical states described above.

### Abnormal network dynamics at 2-weeks correlate with behavioral deficits and recovery in multiple cognitive domains

Finally, we wished to establish whether dynamic FC measures relate to behavioral deficits, and whether, from a clinical standpoint, sub-acute dynamic state measures can improve clinical outcome explanation vis-à-vis static FC.

As expected from previous works[33,43], static FC impairments explain part of the inter-subject behavioral deficits' variance at the sub-acute stage (2 weeks). We verified whether the addition of dynamical functional information to the static FC significantly increases the ability to explain behavioral deficits in terms of generalized linear models (GLM) through a likelihood ratio test, which takes into account the number of regressors. Then, in case of a significant increase, we applied the same test to examine whether the combination of static and dynamic regressors also improves over the model with only dynamical regressors. This second step was aimed at testing whether both the static and the dynamic contributions are jointly needed to explain behavior.

The first analysis uses static and dynamic PC scores to explain overall stroke impairment as measured with the NIH Stroke Scale (NIHSS). The addition of the dynamic PCs significantly increased the explanation of behavioral variability ($R^2$ from 0.38 to 0.52, $\chi^2 = 10.09, df = 3, p = 0.018$) (Fig. 7 left). At the same time, the combined model also outperformed the model with only dynamical regressors ($R^2 = 0.39, \chi^2 = 9.31, df = 1, p = 0.002$), indicating that both contributions were significantly important, with a similar $R^2$. The scatter plot shows the relationship for the combined model (Fig. 7a).

The second analysis examined whether sub-acute static or dynamical FC explains behavioral recovery, measured as the ratio between the difference of behavioral scores at 1 year versus 2 weeks and the absolute value of the behavioral score at 2 weeks. The dynamic FC contribution is relevant in several domains. Specifically, 2-week Dyn-PCs improve model performance for Language recovery ($R^2$ from 0.004 to 0.23, $\chi^2 = 10.71, df = 3, p = 0.013$; $b_1 = 0.91, p = n.s.$; $b_2 = 0.78, p = n.s.$; $b_3 = -1.13, p = 0.018$) and Verbal Memory recovery ($R^2$ from 0.001 to 0.31, $\chi^2 = 12.81, df = 3, p = 0.005$; $b_1 = 0.74, p = n.s.$; $b_2 = 0.29, p = n.s.$; $b_3 = -0.71, p = 0.012$), whereas the 2-week static PC factor is almost irrelevant (Fig. 7b).

In summary, the dynamic measures improve the sub-acute explanation of overall impairment (NIHSS) above static FC. Moreover, dynamic functional measures are suitable to explain future recovery of function of some individual domains. In contrast, static measures were less powerful to explain recovery.

## Discussion

FC in healthy subjects is the result of a sequence of transitions between a set of dynamic functional states (DFSs) alternating in time. These configurations are characterized by specific correlation/anti-correlation patterns of correlation between cortical networks and cortical-subcortical interactions. At the cortical level, the different DFSs reflect

the flexible arrangement of different networks along a single principal gradient of functional organization[53,54], at whose extremes we find, respectively, the visual network and the DMN (SI-Fig. 14). DFS1, the most frequent state in healthy subjects, captures the standard 'static' (time-averaged) task-positive/task-negative pattern separating sensory-motor-attention networks from default/limbic networks. DFS2-5 represent variations of this main pattern, with shifts of some networks along the principal gradient. For instance, DFS2 shows a positive correlation of sensorimotor with cognitive networks (DAN, VAN, CON, FPN), but a relative segregation of the two extremes (visual, DMN). DFS3 shows a positive correlation of visual, sensory-motor, and auditory networks, separate from the correlation of FPN and DMN, with a relative independence of networks laying in-between in correlation space (DAN, VAN, CON) (Figs. 2 and 3). This flexible arrangement determines a changing balance between segregation and integration. Time-resolved FC patterns alternate between states of stronger integration (DFS2 and DFS5) and states of stronger segregation (DFS1 and DFS3). The integration/segregation balance is well captured by static FC metrics affected in stroke: the mean inter-hemispheric FC and whole-brain modularity reflect higher network segregation, whereas abnormally strong DAN-DMN FC reflects higher network integration.

Importantly, we show that cortical networks are flexibly synchronized with two groups of subcortical regions (Fig. 3). One group includes the striatum (caudate, putamen, nucleus accumbens) and anatomically connected globus pallidus and thalamus. The other group includes the hippocampus and amygdala that are part of the limbic system. Cortico-subcortical coordination is not limited to moments of switching between different DFSs, but it is continuously at play, since cortical and subcortical regions show a general simultaneity of dynamical changes (Fig. 4). Most interestingly, the relationship between basal ganglia/thalamus and limbic nuclei seems to be competitive: cortical networks synchronize with either group, but not both at the same time. For instance, in DFS1 DMN shows a positive correlation with limbic regions, and a negative correlation with basal ganglia/thalamus, while the reverse is true in DFS3 (Fig. 3). The striatum/hippocampus competition is also consistent with the emerging role of these regions as waypoints of functional integration or segregation at the whole-brain level[39]. Indeed, striatum, hippocampus and thalamus are part of the topological rich-club[21,55,56], which acts to link specialized large-scale functional systems to ensure high efficiency for information transmission[22,57,58]. In line with previous work that identified the hippocampus as a main region of coordination of cortical networks at rest[20,25–27], we speculate that the hippocampus may play a functional role in facilitating switches between different patterns of cortical activation.

Overall, these findings emphasize the importance of subcortical states in the coordination of cortical dynamics, and in the large-scale network communication and organization[27,40,41]. Hence we suggest that mathematical models aimed at understanding the large-scale functional organization of the brain should include subcortical regions and subcortical-cortical interactions.

A few recent studies investigated dynamic FC changes in stroke. Focusing on the standard deviation of FC links over time, Chen et al.[37] showed that link variability within the motor network is reduced in stroke, while Hu et al.[59] showed link variability in several networks is reduced during the acute stroke stage, and recovers at the chronic stage. Other studies looked at dynamic FC patterns with a DFS approach similar to the one adopted in this work. Bonkhoff et al.[35] characterized DFSs within the motor network, showing that DFS fraction times are different in severe patients/mild patients/controls. Duncan and Small[60] identified a characteristic DFS correlated with post-stroke aphasia severity. Wang et al.[61] compared healthy subjects and stroke patients with midbrain lesions, finding altered DFSs fraction times in patients. Finally, Bonkhoff et al.[36] studied DFSs longitudinally after stroke: they found different DFS fraction times for severe and mild patients, and they showed that the change in symptom severity in the first 3 months post-stroke (NIHSS change), was linked to dynamic connectivity involving DMN components.

Altogether, previous work indicates that (i) stroke patients can exhibit an anomalous preference towards specific DFSs and (ii) dynamic FC anomalies tend to disappear with recovery. Our work confirms and generalizes these findings, by systematically analyzing longitudinal changes of dynamic FC in a relatively large stroke patient cohort including variable lesion sites and deficits in multiple behavioral domains. In addition, by thoroughly analyzing cortical-subcortical interactions and relating dynamic FC changes with lesion topography, our work suggests that subcortical regions play a key role in the altered dynamical balance of the brain after stroke.

Importantly, our analysis highlights the relations between static and dynamic FC changes. Static FC analysis identified well-established stroke anomalies in homotopic connectivity, network modularity, and relative coupling of DAN-DMN intra-hemispheric connectivity. The dynamic FC analysis shows that this static description is the result of an abnormal imbalance among dynamical states in patients, with longer periods and more shifts toward states of integration across cortical networks (DFS2,4), and less frequent states of segregation and strong homotopic connectivity (DFS1 and DFS3) (Fig. 5). These results are generally in agreement with previous findings. Wang et al.[61] identified four DFSs, and showed that patients overexpress (high fraction time) a state with high integration but weak correlations (akin to DFS4 in our study) and underexpress a segregated state with negative correlations between DMN and visual-sensorimotor-attention (akin to DFS1 in our study). Bonkhoff et al.[36] identified three DFSs, one with strong segregation of VIS and SMN from other networks (akin to DFS2 in this work), one with weak correlations (akin to DFS4), and one with anticorrelations between visual-sensorimotor networks and DMN (akin to DFS1). Stroke patients overexpressed the first state: although Bonkhoff et al. characterized this state as a state of anomalous segregation, segregation mostly occurs for the visual/SMN network, while cognitive networks are quite integrated (as in our DFS2).

Static and dynamic FC represent descriptions of the same phenomenon at different time scales as evident in three main results. First, dynamic FC changes were more evident in the subgroup of stroke patients with stronger static impairment. This result is not biased by sampling the same group of subjects as the cut-off for stronger/weaker impairment is determined in a separate sample. Second, both static and dynamic FC deficits are more evident in patients with subcortical lesions disconnecting the basal ganglia and thalamus from cortex (Fig. 6), while patients with milder dynamic FC deficits have more cortical lesions (Fig. 6a). Third, the precise distribution and topography of DFSs is modulated by static FC abnormalities (and vice versa). Indeed, even though our description has emphasized quantitative changes in DFS dynamic properties between healthy control and patients, it is apparent that when we run the analysis separately for the two groups some qualitative differences emerge (SI-Fig. 17). Specifically, the more integrated state DFS2, which is abnormally more frequent in highly impaired patients, looks different in controls and in sub-acute patients. The main difference is a loss of inter-hemispheric connectivity which is the most common abnormality of static FC at the sub-acute stage.

Dynamic FC changes were clearly behaviorally relevant as they improved the explanation of sub-acute impairment measured by the NIHSS, above that provided by static FC. More interestingly, dynamic FC allows acute-to-chronic explanation of the recovery of language and memory scores in contrast to static FC (Fig. 7). These findings unambiguously indicate the potential of dynamic FC to explain deficits, especially for cognitive functions. This is consistent with the notion that functional alterations of brain networks are important for cognitive functions that rely on distributed networks (e.g., memory, attention, language), as compared to visual and motor functions for which structural damage is more sensitive[32].

Dynamic FC recovers over time in parallel to behavior[62], as previously reported for static FC[33]. Behavioral recovery is related to the normalization of the dynamical measures that become similar to those of control subjects, even though stroke patients with different degrees of stroke severity showed different sub-acute dynamic impairments. Our results support the hypothesis that the functional reorganization of brain connectivity after stroke tends to the common goal of regaining a healthy profile, rather than building on compensatory mechanisms.

There are potential limitations to our work. First, several methods exist for the study of FC dynamics[17,19,28], including the variability of temporal correlation patterns[18,63–66], instantaneous phase coherence among brain regions[46,67,68], temporal independent components or modes (TFM)[69], spatial patterns of instantaneous BOLD activity peaks or "co-activation patterns" (CAPs)[70–72], Hidden Markov model-based definition of brain states over time[73–76], and the most recent edge-centric approach to functional connectivity based on edge time series and network bipartitions[77,78]. We opted for the most straightforward approach of a sliding-window temporal correlation followed by eigenvector decomposition and clustering to define a set of spatial connectivity states that continuously activate and deactivate over time. More complex methods to define the connectivity may allow observing a richer variety of dynamical states. However, the sliding-window approach is quite standard and can be easily implemented by different groups of researchers, thus promoting replicability and comparisons across studies. Second, we do not observe a specific "stroke state" activating only in patients. However, this may be due to our methodological choice: concatenating all subjects and conditions before applying the clustering algorithm may decrease the specificity of the sub-acute patients' contribution. We used a single concatenation because having a different clustering for each different condition would have made control/patient comparisons quite difficult. In fact, when we defined DFSs separately in different groups (controls, sub-acute, chronic, mild vs severe patients) (SI-Fig. 17), despite an overall similarity, we did note some differences between controls and sub-acute patients in DFS2 as discussed above. Another possible reason for the absence of a "stroke state" could be the high heterogeneity of patients in terms of lesions and behavioral deficits, as enhanced by the significant dynamical differences between more and less severe stroke patients. Third, to study the relation between FC/dFC and lesions/behavior we made a radical dimensionality reduction step: multivariate information about FC/dFC was effectively condensed in a short array of scalar quantities: three dynamical principal components, summarizing the fraction times, dwell times and transition probabilities of different DFSs; and one static principal component, summarizing the most common pattern of anomalous FC in stroke patients. These scalar quantities cannot be assumed to faithfully represent all possibly relevant aspects of FC and dFC. In principle, a possible alternative to this large dimensionality reduction would have been to use double-multivariate methods such as partial least squares or canonical correlation analysis, but we are unsure whether this would have led to easily interpretable results. Moreover, we did not test for generalization of ridge regression findings to new samples, therefore current findings of ridge regression analyses may be specific to the used subjects' sample. While performing nested cross-validation may enhance robustness of these findings, we believe that conclusive evidence may be obtained only by replicating these findings in an independent subject cohort. Finally, due to the long TR and the large impact of motion scrubbing (on average, 25% of data points are discarded), the amount of data available per subject is limited, which limits the reliability of individual estimates of dynamic FC metrics. Therefore, while our group results indicate that dynamic FC metrics are correlated with stroke severity, their use as individual biomarkers is currently limited.

A key goal of stroke research is to devise novel treatment strategies based on drug therapies[79,80], rehabilitation[81,82], or non-invasive brain stimulation[83,84]. To this aim, the identification of a suitable mathematical model of brain dynamics would be very helpful, not only to explain and explore the patients' empirical data, but also to predict and apply optimal strategies to improve the recovery of behavioral performance, which usually comes in parallel with normalization of FC[85–88]. Our work provides a step toward this direction by discovering the fundamental role of subcortical regions in cortical state dynamics, and by proposing a definition of brain states and a set of measures useful to identify functional abnormalities.

## Methods

This research complies with all relevant ethical regulations. Written informed consent was obtained from all participants in accordance with the Declaration of Helsinki and procedures established by the Washington University in Saint Louis Institutional Review Board. All participants were compensated for their time. All aspects of this study were approved by the Washington University School of Medicine (WUSM) Internal Review Board.

### Imaging

The data used in these analyses are part of the Washington Stroke Cohort[33,38,89]. The database contains patients with first-time stroke, studied 1–2 weeks (mean = 13.4 days, SD = 4.8 days), 3 months, and 12 months after stroke onset. A group of 30 age-matched control subjects was studied twice at an interval of 3 months. All imaging was performed using a Siemens 3T Tim-Trio scanner at the Washington University School of Medicine (WUSM) and a standard 12-channel head coil. The imaging protocol included structural MRI, resting-state blood oxygen dependent level (BOLD) MRI, diffusion MRI, and arterial spin labeling. Structural scans included: (1) a sagittal T1-weighted MP-RAGE (TR = 1950 ms, TE = 2.26 ms, flip angle = 90°, voxel size = 1.0 × 1.0 × 1.0 mm); (2) a transverse T2-weighted turbo spin-echo (TR = 2500 ms, TE = 435 ms, voxel-size = 1.0 × 1.0 × 1.0 mm); and (3) sagittal FLAIR (fluid-attenuated inversion recovery) (TR = 7500 ms, TE = 326 ms, voxel-size = 1.5 × 1.5 × 1.5 mm). Resting-state functional scans were acquired with a gradient echo EPI sequence (TR = 2000 ms, TE = 27 ms, 32 contiguous 4 mm slices, 4 × 4 mm in-plane resolution) during which participants were instructed to fixate on a small white cross centered on a screen with a black background in a low luminance environment. Six to eight resting state (RS) fMRI runs, each including 128 volumes (30 min total), were acquired. Resting-state fMRI pre-processing included (i) regression of head motion, signal from ventricles and CSF, signal from white matter, global signal (ii) temporal filtering retaining frequencies in 0.009–0.08 Hz band; and (iii) frame censoring, FD = 0.5 mm. Finally, the resulting residual time series were projected on the cortical surface of each subject divided into the 324 ROIs developed by Gordon et al.[90], plus 19 subcortical ROIs derived from the FreeSurfer subcortical atlas[91,92]. The original parcellation includes 333 regions, but all regions with <20 vertices (~50 mm²) were excluded, as in previous works[33,43]. We used this parcellation to relate our results with previous works analyzing the same dataset[33,34,38,39,43,88]. Only subjects with at least 180 good frames were considered for the analyses. As a result of the pre-processing, 114 subjects were available at 2 weeks (sub-acute), 80 at 3 months, and 65 at 12 months, 24 and 20 controls at the first and second acquisition, respectively.

### Dynamical functional states (DFSs)

**DFSs definition.** The definition of dynamical functional states (DFSs) required several adjacent frames, thus only recordings with at least 300 TR (600 s) of valid signals (after pre-processing and censoring) were considered valid recordings. Only patients who underwent valid recordings in all three time points (2 weeks, 3 months, 12 months) were

considered in the analysis. We thus considered 20 controls at the first run, 20 controls at the second run, and 47 patients at three time points −in total, $20 + 20 + 47 \times 3 = 181$ sessions. To avoid biases, each session was shortened to 300 TR by excluding high-motion frames and keeping the first 300 good frames. For the clustering step, we concatenated all 181 sessions. The MATLAB-based pipeline for the DFSs' definition consists of 4 steps (Fig. 2): (i) the time course of BOLD response for each subject was divided into 270 overlapping time-windows 30 TR (60 s) long, stepped every 1 TR (2 s). As the optimal choice of the sliding window width is still under debate, in the Supplementary Information (SI-Paragraph 4 and SI-Figs. 6, 7), we compare results obtained with different widths (20, 40, and 50 TR). The z-Fisher transform of Pearson's correlation coefficient among all pairs of regions was computed in each sliding window to estimate the Dynamical Functional Connectivity (DFC); (ii) each DFC matrix was approximated by projecting it on the leading eigenspace defined by the first eigenvector $\mathbf{v}_i$, i.e., by approximating the original DFC matrix with the matrix $\mathbf{v}_i \times \mathbf{v}_i^T$; (iii) the upper triangular part of these rank-one DFC matrices was vectorized and concatenated across windows, subjects, and time points (obtaining a matrix of dimension $48{,}870 \times 4005$, $48{,}870$, and $4005$ being the number of total sliding windows the number of connectivity pairs, respectively), to finally apply a time-wise K-means clustering algorithm[93] with correlation distance and 20 replicates; (iv) Silhouette and Davies-Bouldin algorithms were used to search for the optimal number of DFSs, K. Several choices of K (from 2 to 10) were used for supplementary analyses and comparisons (SI-Fig. 2).

For each DFS, we evaluated three indexes that represent the most common static FC biomarkers observed in stroke, namely: (1) the average homotopic inter-hemispheric connectivity within each network; (2) the average intra-hemispheric connectivity between DAN and DMN regions, as a measure of network integration; (3) the overall Newman's modularity among cortical networks. Similar to ref. 33, we used the code from the Brain Connectivity Toolbox[94], publicly available at sites https://sites.google.com/site/bctnet/, with modules' assignments chosen a priori based on Info-Map community detection in ref. 90. Modularity was calculated at edge densities ranging from 4 to 20%, as suggested in ref. 95, without binarizing and with the symmetric treatment of negative weights. The average modularity across densities was used as the final measure.

All the three metrics were computed in each sliding window, and then sorted across DFSs.

**Dynamic functional states dynamics.** The K-means clustering associated each sliding window to a specific DFS, thus for each subject we obtained a discrete-time series $x(n)$, with $n = 1, \ldots, 270$, where each discrete value (between 1 and K) indicated the active state at that time window. From these time courses, it was possible to evaluate three different dynamical measures for each state, namely the fraction time, the dwell time, and the transition probability (Fig. 1). The fraction time $f_k$ of each DFS is given by the percentage of times during which the state is active:

$$f_k := \frac{\#(x(n) = k)}{270}, \; k = 1, \ldots, K \tag{1}$$

where $\#(a)$ indicates the number of times in which condition $a$ is verified.

The dwell time $\ell_k$ of each DFS measures the average length of periods in which each state remains continuously active. Formally, it is defined as

$$\ell_k := \frac{1}{|L_k|} \sum_{i=1}^{|L_k|} L_k[i] \tag{2}$$

where $L_k$ is the set with cardinality $|L_k|$, and whose elements $L_k[i]$ represent the duration of each period of continuous activity of state $k$:

$$L_k[i] := T_i \text{ such that } x(n), \ldots, x(n + T_i - 1) = k, x(n - 1) \neq k, x(n + T_i) \neq k \tag{3}$$

Finally, the transition probability from DFS $i$ to DFS $j$, $DFSi > j$ is given by the following equation:

$$DFSi > j := \frac{\#(x(n) = i \bigwedge x(n + 1) = j)}{\#(x(n) \neq x(n + 1))} \tag{4}$$

which reflects the ratio between the number of jumps from DFS $i$ to DFS $j$ over the total number of jumps.

Significant differences in terms of dynamical measures were tested across populations (controls, 2-week sub-acute patients, chronic patients at 3 or 12 months) through generalized linear mixed-effects models (GLME), non-parametric permutation tests (when pairs of groups were tested), and one-way Kruskal–Wallis test (for comparisons between more than two groups), to identify abnormal patterns of states dynamics in sub-acute stroke, which might recover after 3 or 12 months.

**Cortical vs subcortical pattern reorganization**
We used the leading eigenvectors of the DFC matrices in each sliding window to define to quantify pattern reorganization in subcortical nuclei and cortical networks during time. Specifically, we considered the $(19 \times 1)$ subvector $\mathbf{v}_{sub}$ of the principal vector $\mathbf{v}_i$, obtained from the 19 entries corresponding to subcortical regions.

The vectors $\mathbf{v}_{sub}$ obtained in different sliding windows were entered as input of a spatial PCA aimed at identifying clusters of subcortical regions that evolve coherently. We thus have $\mathbf{v}_{sub}(t) = P\mathbf{w}_{sub}(t)$ where $P$ is the $(19 \times 2)$ matrix of principal component loadings.

As a measure of overall connectivity for each cortical network, we considered the average of the subvector $\mathbf{v}_{net}$ obtained from the entries of $\mathbf{v}_i$ related to the network regions, in each sliding window.

For each DFS, we computed a measure of connectivity between subcortical and cortical networks (see Fig. 3d). To this aim, we considered the sub-matrix $\mathbf{v}_{net} \times \mathbf{v}_{sub}^T$ of $\mathbf{v}_i \times \mathbf{v}_i^T$, and we projected it onto the principal component space taking $\mathbf{v}_{net} \times \mathbf{v}_{sub}^T P^T$.

In addition, we defined network-wise shifts in connectivity by computing the absolute value of the difference of $\mathbf{v}_{net}$ (for cortical networks) or $P\mathbf{v}_{sub}$ (for subcortical principal components) between two consecutive sliding windows. These shifts followed a heavy tail distribution, with frequent small values and infrequent large shifts. Thus, we binarized the variability time courses (using a threshold of 0.29, corresponding to the 95th percentile) of the subcortical components and of each network.

To test whether subcortical and cortical shifts occurred simultaneously, for each subject separately, we evaluated the observed probability that given a cortical shift in a specific sliding window, a subcortical shift happened in the same sliding window, as follows:

$$P(\text{subcortical shift}|\text{network } i \text{ shift}) = \frac{\#(\text{subcortical shift} \cap \text{network } i \text{ shift})}{\#(\text{network } i \text{ shift})}$$

$$(5.1)$$

We then compared this observed probability with the conditional probability obtained under the null hypothesis of independent processes. Specifically, for each subject, we estimated the probability of a shift as the percentage of ones in the binarized time courses, both for subcortical components ($p_{sub}$) and all cortical networks ($p_{net}$). Under the assumption of independent processes, we evaluated the

conditional probability as follows:

$$P_{H_0}(\text{subcortical shift}|\text{network }i\text{ shift}) = P_{H_0}(\text{subcortical shift}) = p_{sub}$$
(5.2)

Finally, the presence of significant differences in these probability distributions over subjects was assessed by means of non-parametric Wilcoxon rank test.

## Relationship with anatomical lesions and structural disconnections

Structural MRI data acquired from 47 sub-acute stroke patients used for the DFSs analyses were used to measure each lesion's anatomical impact at the voxel level. Lesions were manually segmented on each patient's structural MRI scans using the Analyze software package[96]. The T1-weighted, T2-weighted, and T2-FLAIR scans were used in conjunction to ensure complete lesion delineation. If present, surrounding vasogenic edema was included in the lesion definition for all patients. All segmentations were reviewed by two board-certified neurologists (Maurizio Corbetta-MC- and Alexandre Carter) and were reviewed a second time by MC. The final segmentations were used as binary lesion masks for subsequent processing and analysis steps. Lesion masks were transformed into MNI atlas space using a combination of linear transformations and non-linear warps and were resampled to have isotropic voxel resolution.

The structural disconnectome matrices used in this work were derived in ref. 34 for the same patients' dataset, using the same 324 cortical parcels used in this work and 35 subcortical and cerebellar regions (34 parcels from the automatic anatomical labeling (AAL) atlas[97] that corresponded to different portions of the thalamus, basal ganglia, and cerebellum, and also included 1 region from the Harvard-Oxford Subcortical Atlas that corresponded to the brainstem). Specifically, for each patient, the disconnectome matrix was defined as a square matrix of dimensions $359 \times 359$, where each entry in position $ij$ represented the percentage of streamlines connected regions $i$ and $j$ that were disconnected by the lesion. As described in ref. 34, the template structural connectome was derived from a publicly available diffusion MRI streamline tractography atlas, constructed using data from 842 Human Connectome Project participants[98]. The atlas data were accessed under the WU-Minn HCP open access data use term.

For the computation of structural disconnection matrices, the following software was used: DSI studio 2019 (http://dsi-studio.labsolver.org), FreeSurfer V6 (https://surfer.nmr.mgh.harvard.edu), Connectome workbench v1.5.0 (https://www.humanconnectome.org/software/get-connectome-workbench), GRETNA 22.0 (https://www.nitrc.org/projects/gretna/), Analyze v.12 (https://analyzedirect.com/), Surf Ice v2 (https://www.nitrc.org/projects/surfice), MRIcroGL v1.2.2021 (https://www.nitrc.org/projects/mricrogl).

**Lesion to dynamical patterns regression.** At this point, we wanted to test the relationship of (i) anatomical lesion location and (ii) structural disconnections with the dynamical measures (described by dynamical PCs). Thus, we implemented twice a ridge regression algorithm (RR)[99]: first (i) to link the voxel-wise lesion maps (regressors) to the dynamical PCs scores (dependent variables), once at a time, and second (ii) to link the parcel-wise disconnections matrices (regressors) to the dynamical PCs scores (dependent variables), once at a time.

Specifically, in the lesions-based analysis (i), the regression matrix is a binary matrix $X \in R^{N_s \times N_p}$ ($N_s$ is the number of subjects and $N_p$ is the number of regressors or parameters), whose entry i–j is equal to 1 if voxel j is lesioned in subject i, and it is equal to 0 otherwise. This analysis aimed to assign a weight $\beta_j$ to each voxel, indicating its contribution to the considered dynamical PC values. For instance, a positive (negative) $\beta_j$ indicate that a lesion in that voxel would be probably linked to a positive (negative) value in the considered dynamical PC.

RR adds an L2-normalization term to the ordinary linear regression, to assign small coefficients to unimportant regressors, thus preventing data overfitting, and improving generalization for new data. Specifically, the model weights vector $\beta$ is estimated as:

$$\beta = \left(X^T X + \lambda I\right)^{-1} X^T \mathbf{y}$$
(6)

where $X \in R^{N_s \times N_p}$ is the regressors' matrix described above, $\mathbf{y} \in R^{N_s}$ is the vector containing the (z-scored) scores of the considered dynamical PC, $I \in R^{N_p \times N_p}$ is the identity matrix of dimension $N_p$, and $\lambda \in R$ is the regularization parameter.

Due to computational issues, a dimensionality reduction of matrix $X$ was required before applying the RR. Thus, we applied a PCA on the 147,465 3-mm³ brain voxels, and we considered the first PCs which explained at least 97% of the original variance as regressors $X$ for our RR models. Besides resolving the dimensionality problem, the PCA step also had the purpose to transform the original binary matrix into a set of continuous regressors. $X$ was then z-scored with respect to the whole matrix.

For each of the three RR models (one model for each dynamical PC), the regularization parameter $\lambda$ was optimized by identifying a value within $\left[10^{-5}, 10^5\right]$, with 200 logarithmic steps. Specifically, for each value of $\lambda$, each RR model was trained and tested using a leave-one-out cross-validation loop (LOOCV), which used $47 - 1 = 46$ training data to estimate the model weights and applied them to the left-out patient to explain his behavioral score. The optimal $\lambda$ ($\lambda_{opt}$) value was the one that minimized the prediction error over the training set, and the predictions obtained with $\lambda_{opt}$ were considered as the model regressors.

Model accuracy was assessed through the coefficient of determination $R^2$:

$$R^2 = 1 - \frac{\sum_{i=1}^{N_s}(y_i - \hat{y}_i)^2}{\sum_{i=1}^{N_s}(y_i - y')^2}, \text{where } y' = \frac{1}{N_s}\sum_{i=1}^{N_s}y_i$$
(7)

and $\hat{y}_i$ is the estimated value of $y_i$. The statistical significance of each model was estimated through a permutation test, with $N = 10,000$ iterations. For each iteration, the behavioral scores were randomly permuted across subjects, and the LOOCV with $\lambda$ optimization was used to fit the RR model to the randomized scores. The $p$-value for the observed $R^2$ was defined as the probability of the $R^2$ of the randomized dataset to be larger than the observed $R^2$. Only models with $p$-values < 0.05 were considered statistically able to explain the dynamical scores.

To obtain the optimal set of RR model weights $\beta$, the weights derived from each LOOCV loop at $\lambda_{opt}$ were averaged across the $N_s$ loops. The distribution of weights obtained with the permutation test was used as a null distribution to select the statistically significant weights. Only the $\beta_i$'s that fall at the left or right ends (2.5%) of the tails of the null distribution were considered significant. These selected weights were back-projected to the brain to display a map of the most predictive lesioned voxels. Finally, Gaussian smoothing (variance = 1) and scaling within $[-1, +1]$ was applied to the maps. Only weights higher than 0.05 in absolute values were plotted. To visualize the maps, we used FSLeyes 0.34.2 (https://fsl.fmrib.ox.ac.uk/fsl/fslwiki/FSLeyes). The same algorithm and procedure were used for the disconnections matrices (ii), after that the upper-triangular part of the matrix of each patient was vectorized, obtaining a regressor matrix $X$, whose entry i–j represents the percentage of disconnected streamlines of a specific pair of regions in subject i. The significant weights obtained with RR were back-projected to the parcel-wise matrix, indicating the most predictive pairwise disconnected link.

## Relationship with behavior

**Neuropsychological evaluation.** The same subjects (controls and patients) were also examined at each time point with a battery of neuropsychological tests covering different cognitive domains, such as motor, attention, language, visual, and memory functions as described in refs. [38,43]. Briefly, they included the following tests. Language: Boston Diagnostic Aphasia Examination, nonword reading, stem completion, and animal naming. Motor: Active Range Of Motion, Jamar Dynamometer, nine-hole peg test, Action Research Arm Test, motricity index, and Functional Independence Measures walk test. Attention: Posner visual orienting task, Mesulam symbol cancellation test, and Behavioral Inattention Test Star Cancellation. Memory: Brief Visuospatial Memory Test, Hopkin's Verbal Learning Test, and spatial span. Visual: computerized perimetry. Imaging and behavioral testing sessions were usually performed on the same day. Scores were only recorded for tasks that subjects were able to complete. Dimensionality reduction was performed on the performance data using principal component analysis as described in detail in ref. [38]. Briefly, tasks were first categorized as language, motor, attention, memory, and visual. Next, a PCA was run on each category and the first component was used as a domain score. Finally, patients' behavioral scores were z-scored w.r.t. controls' scores, to highlights behavioral impairments. Of the 47 patients analyzed, the behavioral scores were available for 45 patients (language), 43 patients (motor left and right), 40 patients (attention), 38 patients (memory), and 24 patients (visual) at the sub-acute stage. Similarly, at three months (one year) the following data were available: 45 (41) patients (language), 46 (42) patients (motor), 42 (40) patients (attention), 41 (42) patients (memory), and 28 (28) patients (visual).

In addition to these domain-specific scores, the patients' clinical severity was assessed through the National Institutes of Health Stroke Scale (NIHSS)[100], which includes 15 subtests addressing: level of consciousness (LOC), gaze and visual field deficits, facial palsy, upper and lower motor deficits, limb ataxia, sensory impairment, inattention, dysarthria and language deficits. The total NIHSS was used as an averaged measure of the clinical severity for each patient. This score was available for 40 patients at the sub-acute stage, and for 42 and 47 at three and twelve months, respectively.

To test whether the dynamical measures added some significant information to the static FC in describing the behavioral outcome, for each domain score, and the total NIHSS score, we applied a nested models' comparison test. Specifically, we first estimated the parameters of a Generalized Linear Model (GLM)[101] with the ST as the regressor and each behavioral score as output. Then, we estimated another GLM for the same output, with both ST and the three dynamical PCs scores as regressors. Finally, the Likelihood Ratio Test[102] was used to test if the addition of the dynamical measures were significantly useful to describe the behavioral scores.

In case of a significant increase in performance, we also tested whether the complete (static + dynamic) model outperformed the model with dynamical regressors only, to verify if all the contributions were relevant or not. We used the regressors (static and dynamic) at the sub-acute stage to estimate both the behavioral scores at the sub-acute stage and the behavioral recovery. The recovery was evaluated as the ratio between the difference of behavioral scores at 1 year and 2 weeks, and the absolute value of the score obtained at 2 weeks.

## Statistics and reproducibility

The sample size was determined based on previous works relating to the same dataset[34,38,43]. After pre-processing, 114 subjects were available at 2 weeks (sub-acute), 80 at 3 months, and 65 at 12 months, 24 and 20 controls at the first and second acquisition, respectively. For the implementation of the dynamical functional analysis, only subjects with a sufficient number of good frames (300) after motion scrubbing were considered. Furthermore, we selected only patients who participated to all the three recordings (2 weeks, 3 months, 12 months after stroke). Therefore, 47 patients were considered. The number of control subjects with sufficient frames were 20 during the first visit and 20 during the second visit.

Our statistical analyses are based on common parametric tests (Wilcoxon rank, T-test, F-test for Generalized Linear Mixed Effect Model, likelihood ratio test) or simple non-parametric permutation-based tests which we implemented with customized code. Our code, based on MATLAB 2021a, is available online, as detailed in the "Code availability" section. Statistical tests are described in detail in previous subsections. We always applied correction for multiple comparison whenever testing more than one hypothesis simultaneously.

## Reporting summary

Further information on research design is available in the Nature Research Reporting Summary linked to this article.

## Data availability

Raw neuroimaging and neuropsychological data are publicly available at https://cnda.wustl.edu/data/projects/CCIR_00299 and require controlled access as they contain sensitive patients' data. The person requesting the data must sign a confidentiality agreement provided by Washington University stipulating that they will make no attempt at identifying the patients and that they will use data for research purposes only. Correspondence and requests should be addressed to M.C. (maurizio.corbetta@unipd.it). Source data to reproduce the main figures are provided with this paper.

## Code availability

All custom algorithms used in this work are available at https://github.com/CorbettaLab/Favaretto2022NatComm. Correspondence related to the code should be addressed to C.F. (chiara.favaretto1990@gmail.com) or M.A. (michele.allegra@unipd.it).

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

## Acknowledgements

C.F., M.A., A.B., and M.C. are supported by FLAG-ERA JTC 2017 (grant ANR-17-HBPR-0001). C.F. and M.C. are supported by Fondazione Cassa di Risparmio di Padova e Rovigo (CARIPARO) - Ricerca Scientifica di Eccellenza 2018 – (Grant Agreement number 55403). A.B. has received funding from the European Union's Horizon 2020 Framework Programme for Research and Innovation under the Specific Grant Agreement No. 945539 (Human Brain Project SGA3). M.C. is also supported by MIUR - Departments of Excellence Italian Ministry of Research (MART_ECCELLENZA18_01); Ministry of Health Italy Brain connectivity measured with high-density electroencephalography: a novel neurodiagnostic tool for stroke - NEUROCONN (RF-2008 –12366899); H2020 European School of Network Neuroscience - euSNN, H2020-SC5-2019-2 (Grant Agreement number 869505).

## Author contributions

C.F.: Writing—original draft, software, formal analysis, investigation, writing—review & editing. M.A.: Writing—original draft, investigation, writing—review & editing. G.D.: Writing—review & editing. N.V.M.: Data curation, investigation. J.C.G.: Writing—review & editing. G.L.S.: Writing—review & editing. A.B.: Writing—review & editing. M.C.: Conceptualization, writing—review & editing, supervision.

## Competing interests

The authors declare no competing interests.
