## [Peer Review File · Nature Communications]

Subcortical-cortical dynamical states of the human brain and their breakdown in strokeREVIEWER COMMENTS

Reviewer #1 (Remarks to the Author):

This neuroimaging work aims to assess time-varying functional brain states at rest, focusing on the relationship between subcortical and cortical brain regions. The functional (including behavioural) importance of the detected brain states was assessed by leveraging multimodal clinical (stroke) samples. Overall, the results suggest a key role of subcortical-cortical interactions in the definition of functionally meaningful brain state dynamics. The paper is well written and the results interesting, with potential implications to understand whole-brain principles underpinning recovery from local neural insults. I do, however, have concerns about the aims and message, some methodological decisions, and the stability/validity of the results:

1. Overall message: I found it hard to distil how the present results fit and advance on existing knowledge on brain state dynamics. For example, from the introduction, it is difficult to appreciate what mechanisms supporting the emergence and dissolution of brain states this work tries to elucidate (the sentence starting at line 51 is not sufficient/clear). Moreover, the ambition of studying the emergence of novel 'stroke states' is inconsistent with the adopted method. The discussion section summarises the results well but falls short in explaining how these contribute to the mapping and understanding of brain state dynamics. It is also unclear how the present findings may contribute to the development of new mathematical models of brain dynamics in health and post-stroke.

2. Methods:

2.1. Some of the limitations can be mitigated by performing targeted confirmatory analyses. I encourage the authors to assess the validity of the detected DFS using an alternative methodological approach (e.g., HMM). Moreover, the authors should assess the impact of adopting a superior subcortical parcellation (Tian et al. Nat. Neurosci. 2020; ref. 100) to the definition of the selected DFSs. To be clear, I am not asking the authors to redo all the analyses but to perform a series of specific analyses to boost the confidence in the biological validity of the DFS and the dynamic interplay within/between subcortical and cortical brain regions of interest.

2.2. Frame censoring: A relatively lenient threshold of 0.5 mm (following Power et al.) was adopted to censor volumes contaminated by head motion. Were any additional criteria regarding the censoring of frames used? The censoring of several consecutive frames is likely to impact estimates of time-resolved functional connectivity (via sliding window temporal correlation). Was the number of frames, including the consecutive number, similar across groups? In general, the removal of frames is problematic for measures of brain dynamics and I would like to have some reassurance that this preprocessing step does not explain the results.

2.3. The template used for the functional analyses (343 regions, 19 subcortical) differs from the one adopted to perform the structural analyses (324 cortical parcels and 35 subcortical/cerebellar regions). This seems problematic to interpret the outputs of the RR unequivocally. Can the authors provide evidence that this (unclear) methodological decision does not explain the results (e.g., by calculating DFS on the same "structural" template)?

2.4. I am puzzled by the decision of only considering three dynamic PCs. These PCs only explained 43% of the variability of individual dynamic measures across subjects and states. What was the a priori rationale to only consider three PCs? This rationale should be presented in the text.

3. Results:

Figure 5: I may have missed this, but was there an interaction between group and time (for FO and average life-span)? Testing for a group (CTR/SPC>0/SPC<0) by time (2weeks/3months/1year) interaction seems necessary to claim that acute alterations in brain state dynamics recovered at 3- and 12- months post-stroke (I guess two distinct interactions, 2weeks/3 months and 2 weeks/12 months, could also be justified).

4. Discussion (in addition to what above): "Hence all mathematical models aimed at understanding the functional organisation of the brain shall include subcortical nuclei and connections". This is a very strong claim that is only partially justified by the results, and it is arguably incorrect. I suggest removing this sentence.

Reviewer #2 (Remarks to the Author):

Favaretto et al. present dynamic functional connectivity analyses of longitudinal resting-state fMRI data of 47 ischemic and hemorrhagic stroke patients and 20 healthy controls.

They present several complementary analyses: On the one hand, they examine the interplay of dynamic cortical and subcortical network switches independent of disease. On the other hand, they characterize dynamic functional connectivity of stroke patients in detail. Amongst others, these analyses comprise group comparisons to healthy controls over the first twelve months after stroke, the links between structural lesion and dynamic connectivity patterns and associations to subacute behavior and recovery in the first year post-stroke.

The presented work is altogether quite extensive and methodologically certainly complex. One of the main findings is a particularly intriguing and potentially neuroscientifically relevant one, i.e., that cortical networks synchronized with either limbic or basal ganglia subcortical brain regions and connectivity shifts occurred simultaneously in cortical and subcortical ones. Nonetheless, the current methods may not be described in sufficient detail and additional (sensitivity) analyses may be necessary to ensure the validity of these results, or at least increase the confidence in them. Please find detailed comments and suggestions in the following:

Introduction:

1. The authors could consider mentioning relevant literature on dynamic functional network connectivity (dFNC) analyses in stroke populations (c.f. below for suggestions). These studies already indicate specific dFNC alterations after focal lesions, which, of course, does not preclude that more work needs to be done (like longitudinal evaluations, or establishing links between structural lesion characteristics and dFNC alterations as presented here).

Hu, J., Du, J., Xu, Q., Yang, F., Zeng, F., Weng, Y., Dai, X.J., Qi, R., Liu, X., Lu, G. and Zhang, Z., 2018. Dynamic network analysis reveals altered temporal variability in brain regions after stroke: a longitudinal resting-state fMRI study. *Neural plasticity*, 2018.

Chen, J., Sun, D., Shi, Y., Jin, W., Wang, Y., Xi, Q. and Ren, C., 2018. Alterations of static functional connectivity and dynamic functional connectivity in motor execution regions after stroke. *Neuroscience letters*, 686, pp.112-121.

Duncan, E.S. and Small, S.L., 2018. Changes in dynamic resting state network connectivity following aphasia therapy. *Brain imaging and behavior*, 12(4), pp.1141-

1149.

Bonkhoff, A.K., Espinoza, F.A., Gazula, H., Vergara, V.M., Hensel, L., Michely, J., Paul, T., Rehme, A.K., Volz, L.J., Fink, G.R. and Calhoun, V.D., 2020. Acute ischaemic stroke alters the brain's preference for distinct dynamic connectivity states. *Brain*, 143(5), pp.1525-1540.

Bonkhoff, A.K., Schirmer, M.D., Bretzner, M., Etherton, M., Donahue, K., Tuozzo, C., Nardin, M., Giese, A.K., Wu, O., D. Calhoun, V. and Grefkes, C., 2021. Abnormal dynamic functional connectivity is linked to recovery after acute ischemic stroke. *Human brain mapping*, 42(7), pp.2278-2291.

Wang, Y., Wang, C., Miao, P., Liu, J., Wei, Y., Wu, L., Wang, K. and Cheng, J., 2020. An imbalance between functional segregation and integration in patients with pontine stroke: A dynamic functional network connectivity study. *NeuroImage: Clinical*, 28, p.102507.

Methods/Results:

2. My main concern relates to the censoring preprocessing step. In general, 300 TRs, as utilized here, have certainly proven to be sufficient for dFNC analyses. However, given that each subject had six to eight scans with 128 TRs and therefore 768 or 1,024 TRs, those 300 TRs represent less than 50%? I am assuming all scans were shortened to 300TRs or did subjects have varying scan lengths? In what way were censored frames handled? Were there varying numbers of frames per window or some windows even empty in subjects? In what way were the multiple scans sessions integrated?

Head motion control has been shown to be very essential for dFNC analyses, but as Lurie et al (2020) point out, while there are sporadic positive reports (Laumann et al, 2017), censoring can interfere with robust dFNC estimation (Hutchinson et al, 2013; Zalesky et al, 2014). It is of prime importance for dFNC analyses to retain the correct temporal relationship between frames. Altogether, I currently find it difficult to evaluate the validity of any of the downstream dFNC analyses (e.g., do any of the observed group differences between healthy controls and patients arise due to varying censoring patterns?). The authors might consider conducting sensitivity analyses without censoring and other strategies for motion control, e.g., the inclusion of motion parameters as covariates during dFNC estimation.

Hutchison, R.M., Womelsdorf, T., Allen, E.A., Bandettini, P.A., Calhoun, V.D., Corbetta, M., Della Penna, S., Duyn, J.H., Glover, G.H., Gonzalez-Castillo, J. and Handwerker, D.A., 2013. Dynamic functional connectivity: promise, issues, and interpretations. *Neuroimage*, 80, pp.360-378.

Zalesky, A., Fornito, A., Cocchi, L., Gollo, L.L. and Breakspear, M., 2014. Time-resolved resting-state brain networks. *Proceedings of the National Academy of Sciences*, 111(28), pp.10341-10346.

Laumann, T.O., Snyder, A.Z., Mitra, A., Gordon, E.M., Gratton, C., Adeyemo, B., Gilmore, A.W., Nelson, S.M., Berg, J.J., Greene, D.J. and McCarthy, J.E., 2017. On the stability of BOLD fMRI correlations. *Cerebral cortex*, 27(10), pp.4719-4732.

- this study reports a positive effect of censoring in the context of dFNC analyses, but the authors retained only those segments of data with more than 5 contiguous frames and kept 72.5% of their data in total

Lurie, D.J., Kessler, D., Bassett, D.S., Betzel, R.F., Breakspear, M., Kheilholz, S., Kucyi, A., Liégeois, R., Lindquist, M.A., McIntosh, A.R. and Poldrack, R.A., 2020. Questions and controversies in the study of time-varying functional connectivity in resting fMRI. *Network Neuroscience*, 4(1), pp.30-69.

3. Clustering: Taken from the results section: "To avoid one group dominating the other in the following clustering step, we equalized the number of controls and patients (since patients at different time points are considered different groups)." How many subjects were thus considered for the clustering step? All 20 + 20 controls and then how many patients per time point? Is it that you excluded patients simply for the clustering step and then later on assigned the excluded patients' windows to specific DFSs that were derived from the harmonized group?

4. Dynamic Functional States dynamics: Consider pointing out that "frequency of occurrence" and "average lifespan" correspond to the more commonly used terms "fraction time" and "dwell time" to increase the ease with which your results can be compared to other work.

It would furthermore be helpful, if you added a description of the split into "mild and severe FC impairment" groups, and how this FC impairment was defined, to the methods section. The split based on the FC impairment criterion, as described in the supplement, does not occur to be very straightforward to me, but may be fine from a methods perspective. With respect to any clinical relevance of findings, it might however be interesting to state the behavioral outcome averages per FC defined group. Are the patients with severe FC impairment also the ones that have the highest symptom load post-stroke?

As a side note: To reduce ambiguity, It might be good to go through the entire manuscript and ensure that the groups are always referred to as high low "FC impairment" and not simply high/low "impairment", as it might be confused with actual behavioral impairment.

Could you also describe more in detail which tests were used to examine which group differences? In what way did you incorporate mathematically that these were measurements from the same subjects over time? Did you check whether your input variables to the GLME model (a 2-way ANOVA?) were correlated? In my experience, group differences in derived dFNC measures, like fraction and dwell time, are more commonly tested in 1-way ANOVAs, i.e., separately for each state and not additionally across states in a 2-way ANOVA, as fraction and dwell times of different states are usually highly correlated (c.f., Fu et al. 2019).

Fu, Z., Caprihan, A., Chen, J., Du, Y., Adair, J.C., Sui, J., Rosenberg, G.A. and Calhoun, V.D., 2019. Altered static and dynamic functional network connectivity in Alzheimer's disease and subcortical ischemic vascular disease: shared and specific brain connectivity abnormalities. *Human brain mapping*, 40(11), pp.3203-3221.

5. Cortical vs. subcortical pattern reorganization: I find this analysis particularly intriguing. However, given the novelty of this approach, I would suggest adding more details describing the methodological steps in the actual methods/results sections. It would for example be good to mention the thresholding step that leads to the collection of subcortical regions in the main text (as it's only mentioned in Figure 3 so far?). Otherwise, it appears confusing why the subcortical regions are combined to "clusters" (as PCA-weights are continuous). How was the threshold of 0.2 chosen? Is the brainstem assigned to both subcortical clusters?

Additionally: It would seem more straightforward to me, if the organization of cortical

and subcortical regions into networks/clusters was performed similarly. At the moment, cortical regions are organized into a priori defined networks, while subcortical networks are estimated from the dynamic connectivity data at hand. The authors could consider discussing what effect this methodological difference might have had, e.g. in the Limitations section.

Shifts in cortical connectivity per network are evaluated based on changes between connectivity *averages* for each network and window. Is this the same for subcortical clusters, i.e. the connectivity of all within-cluster connections was averaged? Or was some PCA-weight used?

How was the similarity between cortical and subcortical networks/clusters determined? Given Figure 3D, it's likely the average connectivity between cortical network and subcortical cluster?

Does Figure 4 exemplarily display the time courses of two individual subjects? How were these excerpts chosen?

Were there any group differences in these shift patterns between stroke patients and healthy controls?

6. Relationship with anatomical lesions and structural disconnections: Great idea! In view of the double-multivariate nature of the structural vs. functional data, it might be an optimal use case for a method that is optimized for such a scenario, such as CCA or partial least square regression (c.f., Wang et al, 2020)?

In case the current approach is kept, I would recommend to perform the hyperparameter optimization in a nested cross-validation loop, as the performance estimates can be too optimistic otherwise (c.f., Varoquaux et al, 2017).

Additionally, consider describing how the dynamic PCAs were derived in the main text (currently mostly found in Figure 1?).

Were the beta-weights corrected for multiple comparisons?

Wang, H.T., Smallwood, J., Mourao-Miranda, J., Xia, C.H., Satterthwaite, T.D., Bassett, D.S. and Bzdok, D., 2020. Finding the needle in a high-dimensional haystack: Canonical correlation analysis for neuroscientists. *NeuroImage*, 216, p.116745.

Varoquaux, G., Raamana, P.R., Engemann, D.A., Hoyos-Idrobo, A., Schwartz, Y. and Thirion, B., 2017. Assessing and tuning brain decoders: cross-validation, caveats, and guidelines. *NeuroImage*, 145, pp.166-179.

7. Relationship with behavior: I would suggest describing more in detail in what way FC (and dFNC) information was integrated in these analyses. Did you only use the first two FC PCA-components that were also used to define the split in two $SPC > 0$ and < 0 groups? If so, it would be good to render it more apparent that these analyses compare two very specific aspects of FC and dFNC data (and findings might or might not hold for other representations of FC and dFNC information). Additionally, consider changing your language to "explain" (instead of "predict") throughout the entire manuscript, as no prediction performance per se was tested (analyses were performed in-sample, weren't they?)

Discussion:

8. Two previous dFNC in stroke studies (Bonkhoff et al., 2020, 2021) actually found an increase in network segregation in case of severe stroke and an increase in network integration in case of only moderate motor impairment. These findings of an increased network segregation are in line with the ones obtained from healthy subjects after cast-induced motor inactivity of the upper limb (Newbold et al., 2020). The authors could

consider discussing these previous findings with respect their own ones (previous studies have for example defined groups of stroke patients based on clinical symptoms post-stroke and not static functional connectivity).

Newbold, D.J., Laumann, T.O., Hoyt, C.R., Hampton, J.M., Montez, D.F., Raut, R.V., Ortega, M., Mitra, A., Nielsen, A.N., Miller, D.B. and Adeyemo, B., 2020. Plasticity and spontaneous activity pulses in disused human brain circuits. *Neuron*, 107(3), pp.580-589.

9. Limitation section: Consider discussing differences in the choice of exact methodology (many other dFNC papers would more commonly employ tapered windows, estimate dynamic connectivity via the precision matrix (Prete et al., 2017), use the l1-norm for k-means clustering (Aggarwal et al., 2001) and also perform more preprocessing steps like despiking). Would we expect any differences if analyses were repeated varying these aspects?)

Prete, M.G., Bolton, T.A. and Van De Ville, D., 2017. The dynamic functional connectome: State-of-the-art and perspectives. *Neuroimage*, 160, pp.41-54.

Aggarwal, C.C., Hinneburg, A. and Keim, D.A., 2001, January. On the surprising behavior of distance metrics in high dimensional space. In *International conference on database theory* (pp. 420-434). Springer, Berlin, Heidelberg.

Figures:

10. Please increase the font size to ensure all captions are legible.

Additional points to potentially consider:

11. What does dynamic connectivity tell us that static connectivity did not?

12. It would be good to change "acute" to "subacute" throughout the entire manuscript (c.f., Bernhardt et al., 2017).

Bernhardt, J., Hayward, K.S., Kwakkel, G., Ward, N.S., Wolf, S.L., Borschmann, K., Krakauer, J.W., Boyd, L.A., Carmichael, S.T., Corbett, D. and Cramer, S.C., 2017. Agreed definitions and a shared vision for new standards in stroke recovery research: the stroke recovery and rehabilitation roundtable taskforce. *International Journal of Stroke*, 12(5), pp.444-450.

Reply to reviewer #1

This neuroimaging work aims to assess time-varying functional brain states at rest, focusing on the relationship between subcortical and cortical brain regions. The functional (including behavioural) importance of the detected brain states was assessed by leveraging multimodal clinical (stroke) samples. Overall, the results suggest a key role of subcortical-cortical interactions in the definition of functionally meaningful brain state dynamics. The paper is well written and the results interesting, with potential implications to understand whole-brain principles underpinning recovery from local neural insults.

- We thank the Reviewer for her/his accurate reading of our manuscript, appreciation of our work and valuable comments.

I do, however, have concerns about the aims and message, some methodological decisions, and the stability/validity of the results

- In this revised version, we strived to better convey the main messages of our work, clarify the rationale of our methodological choices, and strengthen our findings showing their robustness to potential confounders.

1. *Overall message: I found it hard to distill how the present results fit and advance on existing knowledge on brain state dynamics. For example, from the introduction, it is difficult to appreciate what mechanisms supporting the emergence and dissolution of brain states this work tries to elucidate (the sentence starting at line 51 is not sufficient/clear).*

- Our work does not provide mechanistic insight on how brain states emerge and dissolve. However, our findings lend support to the view that subcortical regions play a key role in the process in contrast to dominant models emphasizing cortical hubs. We clarify this point in the introduction (lines 55-62):

“The mechanisms controlling the large-scale temporal coordination of infra-slow activity are unclear, particularly whether specific regions play a leading role in orchestrating global changes in connectivity patterns. A leading hypothesis is that shifts in brain states at rest, or during tasks, depend on highly interconnected cortical regions (hubs), e.g., precuneus, posterior cingulate cortex, and lateral prefrontal cortex, that flexibly interact at different points in time with different networks^{30,44,74}. However, recent studies have also shown hubs in subcortical regions (basal ganglia³⁴, thalamus^{33,34}, hippocampus^{29–32}). Whether cortical synchronization also relies on subcortical regions is still poorly known.”

Moreover, the ambition of studying the emergence of novel ‘stroke states’ is inconsistent with the adopted method.

- We agree with this comment. We removed reference to ‘novel stroke states’ in the introduction.

The discussion section summarizes the results well but falls short in explaining how these contribute to the mapping and understanding of brain state dynamics.

- Our manuscript does not propose a new methodology to identify brain states. nor does it discuss new mechanisms for the formation of brain states. However, our results suggest that including subcortical regions in the analysis is crucial for a correct identification and

interpretation of dynamical states. In the revised manuscript, we tried to better contextualize our work and clarify where it improves over previous literature.

It is also unclear how the present findings may contribute to the development of new mathematical models of brain dynamics in health and post-stroke.

- In the revised Discussion, we only make a cursory reference to mathematical models of brain activity (lines 405-407), as our findings do not *per se* imply a refinement of such models. However, they do imply that inclusion of subcortical regions is important especially in the case of lesions that affect cortico-subcortical interactions.

“Hence we suggest that mathematical models aimed at understanding the large-scale functional organization of the brain should include subcortical regions and subcortical-cortical interactions.”

2.1. Some of the limitations can be mitigated by performing targeted confirmatory analyses. I encourage the authors to assess the validity of the detected DFS using an alternative methodological approach (e.g., HMM).

- We do not think that the specific topography of different DFSs is the key point of our work. The DFS landscape will likely not be strongly conserved across different methods. It is unlikely that we would find the same states with, e.g., Hidden Markov Models, or Switching Linear Dynamical Systems. The DFS landscape may substantially change even if we maintain the same methodology but vary some details of the implementation, such as employing tapered windows, or using the L1-norm for K-means. In fact, using the same methodology and only changing the number of states (e.g., $K=2-10$) we obtain a quite different set of DFSs (see SI-Figure 2).

In general, we do not think that the major insights of our work (or other works using DFSs) is strongly tied to the specific DFSs detected. DFSs yield an effective low-dimensional decomposition of the space of windowed FC, which proves very useful to analyze group differences in dynamic FC. These group differences are what matters most in our analyses. For instance, we observed that severe stroke patients ($SPC > 0$) “over-express” dynamical states with low inter-hemispheric integration (such as DFS2 and DFS4). This is a relevant finding, which we expect to emerge even if a different DFS decomposition is used. This can be well exemplified by comparing results for $K=5$ (the one reported in the main analysis) and $K=10$. The set of DFS for $K=5$ cannot be immediately and trivially related to those for $K=10$ (See SI-Figure 2). However, we can sort the states for $K=10$ based on the total homotopic FC (from highest to lowest), as in Fig. R1a. By looking at fraction times, we observe that severe patients ($SPC > 0$) over-express the state(s) with low homotopic FC (Fig. R1b). This is not true for mild patients ($SPC < 0$). Hence, this feature is robustly observed across different DFS decompositions. This analysis is now reported in Supplementary paragraph S6 and SI-Figure 9.

Let us further stress that another key finding – such as the fact that cortical and subcortical FC shifts are simultaneous - does not depend on the chosen set of DFS.

Figure R1 (a) Total homotopic FC for DFSs obtained with K=10 **(b)** Fraction times for DFSs obtained with K=10 (sorted according to descending homotopic FC)

2.2. Moreover, the authors should assess the impact of adopting a superior subcortical parcellation (Tian et al. Nat. Neurosci. 2020; ref. 100) to the definition of the selected DFSs. To be clear, I am not asking the authors to redo all the analyses but to perform a series of specific analyses to boost the confidence in the biological validity of the DFS and the dynamic interplay within/between subcortical and cortical brain regions of interest.

- We thank the Reviewer for this comment. We fully agree on the interest of considering a superior subcortical parcellation. Therefore, we performed additional analyses with Tian et al.'s subcortical parcellation (Nat. Neurosci. 2020), and verified the robustness of three main findings: 1) the 'antagonistic' dynamics of basal ganglia vs limbic regions, represented by two anticorrelated principal components of subcortical dynamic FC; 2) the observation that different DFS are associated with different patterns of cortical/subcortical interactions, as shown by different patterns of connectivity between the main subcortical clusters and cortical networks; and, 3) the coordination between cortical and subcortical dynamics, as shown by simultaneous cortical/subcortical FC shifts.

Tian et al. (Nat. Neurosci. 2020) provide four subcortical parcellations with increasing levels of resolution (16, 32, 50, or 54 regions respectively). We limited our analysis to the lowest (16 regions) and highest (54 regions) resolution parcellations. Results are presented in fig. R2 and R3, respectively.

We recomputed the PCs of the leading eigenvector time courses of subcortical regions in the 16-region parcellation by Tian et al. The first PC, which explains 32% of the variance, loads strongly on basal ganglia (caudate, putamen, globus pallidus) and anterior thalamus, and weakly on posterior thalamus and nucleus accumbens; the second PC, which explains 18% of the variance, loads strongly on limbic regions (hippocampus and amygdala) and weakly on posterior thalamus (fig. R2a). This result broadly agrees with previous findings obtained with the FreeSurfer parcellation: subcortical regions roughly split into a 'basal ganglia' and a 'limbic' cluster. Qualitatively, the only main difference between results in the two parcellations is related to the thalamus. While in the FreeSurfer parcellation the thalamus essentially grouped with the basal ganglia, the new parcellation yields a more nuanced picture, hinting at a division between different parts of the thalamus: the anterior portion of the thalamus groups with the basal ganglia, whereas the posterior portion cannot be clearly affiliated to either of the two clusters (basal ganglia/limbic).

We then performed an analogous analysis with the 54-region parcellation by Tian et al. The first PC, which explains 31% of the variance, loads strongly on caudate, putamen, anterior globus pallidus, anterior thalamus, and weakly on posterior thalamus, posterior globus pallidus and nucleus accumbens. The second PC, which explains 11% of the variance, loads strongly on limbic regions (hippocampus and amygdala), posterior thalamus, some parts of the anterior thalamus, and weakly on other parts of the anterior thalamus (fig. R3). These results strengthen the picture obtained in the 16-region parcellation, but show finer splits within the thalamus, with some portions of the anterior thalamus being associated with the first PC, others with the second PC. In summary, the results in the two parcellations confirm the split between basal ganglia and limbic regions, yielding the additional insight that the thalamus is functionally subdivided into regions showing differential association with the two main clusters.

We checked whether different DFS are associated with different patterns of cortical/subcortical interactions. Fig. R2 shows the average connectivity between the two subcortical PCs and each cortical network in each DFS. As previously discussed, we think that the most important point is not validating the specific DFS found (the number and shape of each DFS can depend on details of the analysis), but rather the global picture of FC dynamics emerging from DFS analysis (including, chiefly, the interplay between cortical and subcortical regions). Therefore, in our new analyses, we did not repeat the clustering step to find new DFS. We only observed whether the old DFSs (obtained using the FreeSurfer parcellation of subcortical regions) correspond to different patterns of cortical/subcortical interaction in the new subcortical parcellation by Tian et al. We used the previous assignment to one of five DFSs and recomputed the connectivity between subcortical PCs and cortical networks in DFSs, using the new subcortical PCs obtained with the parcellation by Tian et al. Results are in qualitative agreement with previous results. Each DFS is characterized by a different set of cortical and cortical-subcortical interactions. Congruently with previous analyses, DFS1 is characterized by a positive correlation between DMN and the limbic cluster, which in turn is negatively correlated with sensory-motor-attention networks. DFS3 is characterized by a negative correlation between DMN and the limbic cluster, which in turn is positively correlated with sensory-motor-attention networks (conversely, the basal ganglia cluster is negatively correlated with DMN and positively correlated with sensorimotor networks). In DFS4 all subcortical regions show positive correlation. Similar evidence is obtained in the 54-region parcellation [not shown].

Finally, we repeated the analysis on the temporal coordination of changes in cortical and subcortical connectivity using the subcortical parcellations by Tian et al. We evaluated connectivity shifts, defined as connectivity differences between pairs of consecutive sliding windows, separately for cortical and subcortical regions. We defined connectivity jumps when a large connectivity difference occurred (0.29, corresponding to the top 5% values), and tested the simultaneity of cortical and subcortical reorganization by comparing the probability that cortical and subcortical jumps occur simultaneously. For all networks, we found that the observed conditioned probability was significantly larger than the probability under the null hypothesis (all $p < 10^{-40}$, Bonferroni corrected for 9 networks).

These results are now reported in SI paragraph S7 and SI-figs. 10-11, and briefly mentioned in the Results section (lines 196-214) after the discussion of cortical/subcortical coordination:

“Importantly, the observed coordination between cortical and subcortical dynamics does not depend on the specific subcortical parcellation used. We replicated our original analyses (based on the Freesurfer parcellation) with a more recent subcortical parcellation (Tian et al. 2020). Tian et al. (Nat. Neurosci. 2020) developed four subcortical parcellations with increasing levels of resolution (16, 32, 50, or 54 regions respectively). We limited our analysis to the lowest (16 regions) and highest (54 regions) resolution parcellations. Detailed results are presented in the Supplementary Information (**SI paragraph S7 and SI-Fig. 10÷11**). The choice of parcellation did not influence our three main findings: 1) the ‘antagonistic’ dynamics of basal ganglia vs limbic regions, represented by two anticorrelated principal components of subcortical dynamic FC; 2) the observation that different DFS are associated with different patterns of cortical/subcortical interactions, as shown by different patterns of connectivity between the main subcortical clusters and cortical networks; and, 3) the coordination between cortical and subcortical dynamics, as shown by simultaneous cortical/subcortical FC shifts. Qualitatively, the main difference between results in the two parcellations is related to the thalamus. While in the Freesurfer parcellation, used in the original analysis, the thalamus essentially grouped with the basal ganglia, the new parcellation yields a more nuanced picture, hinting at a functional split between different parts of the thalamus: the anterior portion of the thalamus groups with the basal ganglia, whereas the posterior portion cannot be clearly affiliated to either of the two clusters (basal ganglia/limbic). A fine-grained analysis of the relation between thalamic nuclei and DFS is left for future work.”

Figure R2 (a) projection of the first three PCs of subcortical FC dynamics on each subcortical region, using the 16-region subcortical parcellation by Tian et al. (b) Connectivity between cortical regions and the first two PCs of subcortical FC dynamics, using the 16-region parcellation by Tian et al. (c) Cumulative density function of the conditioned probability of subcortical connectivity reorganization, given a cortical connectivity reorganization, using the 16-region subcortical parcellation by Tian et al. Each colored line relates to a different cortical network. The black line shows the cumulative density function under the null hypothesis of independence between cortical and subcortical changes.

Figure R3 Projection of the first two PCs of subcortical FC dynamics on each subcortical region, using the 54-region subcortical parcellation by Tian et al.

2.3. *Frame censoring: A relatively lenient threshold of 0.5 mm (following Power et al.) was adopted to censor volumes contaminated by head motion. Were any additional criteria regarding the censoring of frames used? The censoring of several consecutive frames is likely to impact estimates of time-resolved functional connectivity (via sliding window temporal correlation). Was the number of frames, including the consecutive number, similar across groups? In general, the removal of frames is problematic for measures of brain dynamics and I would like to have some reassurance that this preprocessing step does not explain the results.*

- We agree with the reviewer that frame censoring might potentially affect our results, yielding a spurious contribution to the observed patient-control differences. Therefore, we performed control analyses to check the impact of censoring. We recomputed dynamic measures (fraction times, dwell times and transition probabilities) with different censoring thresholds, a much more stringent threshold of 0.25, as well as a more liberal threshold of 0.75 (to facilitate comparison, dynamic measures were computed on the same set of DFS previously identified with a censoring threshold of 0.5). Results are shown in Fig. R4. Qualitatively, we did not observe important differences when varying the threshold. Quantitatively, among the three types of dynamical measures, transition probabilities were more strongly affected by the change in the threshold. However, fraction times were very robust. In particular, the relevant group differences identified in the main text did not depend on the threshold chosen. This analysis is now reported in SI paragraph S5 and SI-Figure 8.

Fig. R4 Fraction times and dwell times obtained with two different censoring thresholds: a more stringent threshold of 0.25 (A), and a more liberal threshold of 0.75 (B).

2.4. The template used for the functional analyses (343 regions, 19 subcortical) differs from the one adopted to perform the structural analyses (324 cortical parcels and 35 subcortical/cerebellar regions). This seems problematic to interpret the outputs of the RR unequivocally. Can the authors provide evidence that this (unclear) methodological decision does not explain the results (e.g., by calculating DFS on the same “structural” template)?

- We agree that it would have been more elegant to use the same parcellation for both the structural and the functional data. We used the structural disconnection matrices in a 324+35 parcellation because they had been already computed in two previous papers [Griffis, Joseph C., et al. "Structural disconnections explain brain network dysfunction after stroke." *Cell reports* 28.10 (2019): 2527-2540; Griffis, Joseph C., et al. "Damage to the

shortest structural paths between brain regions is associated with disruptions of resting-state functional connectivity after stroke." *NeuroImage* 210 (2020): 116589].

However, we do not think that using two different parcellations is problematic *per se* to interpret the ridge-regression results. In fact, our analysis would be entirely meaningful even if we used two widely different sets of regions (let alone two different parcellations). For instance, one could well compute a set cortical DFS (using e.g., the 324 parcellation), and restrict analysis on *subcortical* structural data, to see whether subcortical lesions have an impact on cortical dynamics.

2.5. *I am puzzled by the decision of only considering three dynamic PCs. These PCs only explained 43% of the variability of individual dynamic measures across subjects and states. What was the a priori rationale to only consider three PCs? This rationale should be presented in the text.*

- The main rationale behind this choice is that each of the subsequent principal components (from the fourth onwards) was explaining only a small fraction (<5%) of the total variance. This is now mentioned (lines 298-300):
“We focus on these components since each of the subsequent principal components (from the fourth onwards) explains only a small fraction (<5%) of the total variance.”

3. Results:

Figure 5: I may have missed this, but was there an interaction between group and time (for FO and average life-span)? Testing for a group (CTR/SPC>0/SPC<0) by time (2weeks/3months/1year) interaction seems necessary to claim that acute alterations in brain state dynamics recovered at 3- and 12- months post-stroke (I guess two distinct interactions, 2weeks/3 months and 2 weeks/12 months, could also be justified).

- In our analysis we did not perform a two-way ANOVA with group (CTR/SPC>0,SPC<0) and time as factors. This is mainly because the ‘time’ factor is different for patients and controls. For patients we have recordings at *three* time points, representing different clinical phases. For controls, we have recordings at *two* time points, three months apart that are conceptually interchangeable. Therefore, we performed three separate ANOVAs, considering 1) patients at the sub-acute (2 weeks) stage vs controls (at both time points), 2) patients at the early (3 months) chronic stage vs controls (at both time points) 3) patients at the late (12 months) chronic stage vs controls (at both time points).

Nevertheless, we understand the reviewer’s point of view and we tried to overcome the lack of multiple recordings in controls through the following procedure: we first z-scored the patients’ data at each of the three time points w.r.t. to the controls’ data (averaged over the two recordings), and then we applied a multi-way ANOVA to the z-scored patients’ data. To be more precise, we first computed the mean and the standard deviation of the fraction time of each DFS in the control population (averaging over the two time points), then, we divided the DFSs fraction times of each patient at each time point by the controls’ standard deviation after having removed the controls’ mean. This yields a z-scored measure indicating how much each patient’s measure at each time point differs from the controls’ average distribution.

As patients with severe and mild damage have very different abnormal patterns in the DFSs dynamics, to test recovery of fraction time, we considered the two groups separately and we focused on the subset of DFSs with abnormal fraction times at the acute stage, for each

group. Thus, we tested the time x DFS interaction with two separate 2-Way ANOVAs with repeated measures on the z-scored fraction times:

- for severe patients: time (2w/3m/1y) X DFS (1, 2, 3, 4)
- for mild patients: time (2w/3m/1y) X DFS (2, 5)

As suggested by the reviewer, we can speak of recovery when a significant interaction between time and DFS is found. From these analyses we found a significant interaction between time and DFS for severe patients ($F=2.43$, $p=0.031$) (**Fig. R5**), and the absence of any interaction for mild patients ($F=0.36$, $p=0.70$), thus suggesting that more severe abnormalities at the acute stage are more likely to be recovered with time (in line with previous works [Ref. 85, Ramsey, L. E. *et al.*, *Nat. Hum. Behav.* (2017)]). Specifically, for severe patients, we found a significant main effect of time in DFS2 ($F=3.84$, $p=0.031$) and DFS3 ($F=3.72$, $p=0.035$) and marginally in DFS1 ($F=3.26$, $p=0.051$).

The absence of significant time x DFSs interaction in mild patients, together with the absence of significant difference in fraction time between mild patients and controls at 3 month and 1 year after stroke onset (main text) can be interpreted as a confirmation of the mild level of abnormalities of connectivity dynamics for this group of patients at the acute stage.

Fig. R5 Mean and standard deviation of fraction time (z-scored w.r.t. to control subjects) in severe patients at 2 weeks, 3 months and 1 year.

4. Discussion (in addition to what above): “Hence all mathematical models aimed at understanding the functional organisation of the brain shall include subcortical nuclei and connections”. This is a very strong claim that is only partially justified by the results, and it is arguably incorrect. I suggest removing this sentence.

- We clarified this sentence in the revised version of the manuscript. We think that our findings show that dynamic FC strongly depends co-varies with subcortical-cortical interactions that shall be considered.

Reply to Reviewer #2

Favaretto et al. present dynamic functional connectivity analyses of longitudinal resting-state fMRI data of 47 ischemic and hemorrhagic stroke patients and 20 healthy controls. They present several complementary analyses: On the one hand, they examine the interplay of dynamic cortical and subcortical network switches independent of disease. On the other hand, they characterize dynamic functional connectivity of stroke patients in detail. Amongst others, these analyses comprise group comparisons to healthy controls over the first twelve months after stroke, the links between structural lesion and dynamic connectivity patterns and associations to subacute behavior and recovery in the first year post-stroke.

- We thank the reviewer for this nice summary of our work, and her/his accurate reading of the manuscript, appreciation, and valuable comments.

The presented work is altogether quite extensive and methodologically certainly complex. One of the main findings is a particularly intriguing and potentially neuroscientifically relevant one, i.e., that cortical networks synchronized with either limbic or basal ganglia subcortical brain regions and connectivity shifts occurred simultaneously in cortical and subcortical ones.

- We thank the Reviewer for recognizing the value of our work and potential interest to the neuroscience community.

Nonetheless, the current methods may not be described in sufficient detail and additional (sensitivity) analyses may be necessary to ensure the validity of these results, or at least increase the confidence in them.

- We hope that this revised version addresses the Reviewer's concerns. The paper is stronger as we show that the results do not depend on motion frame censoring, the specific number of DFSs, and the specific subcortical parcellation adopted.

Introduction:

1. The authors could consider mentioning relevant literature on dynamic functional network connectivity (dFNC) analyses in stroke populations (c.f. below for suggestions). These studies already indicate specific dFNC alterations after focal lesions, which, of course, does not preclude that more work needs to be done (like longitudinal evaluations, or establishing links between structural lesion characteristics and dFNC alterations as presented here).

Hu, J., Du, J., Xu, Q., Yang, F., Zeng, F., Weng, Y., Dai, X.J., Qi, R., Liu, X., Lu, G. and Zhang, Z., 2018. Dynamic network analysis reveals altered temporal variability in brain regions after stroke: a longitudinal resting-state fMRI study. Neural plasticity, 2018.

Chen, J., Sun, D., Shi, Y., Jin, W., Wang, Y., Xi, Q. and Ren, C., 2018. Alterations of static functional connectivity and dynamic functional connectivity in motor execution regions after stroke. Neuroscience letters, 686, pp.112-121.

Duncan, E.S. and Small, S.L., 2018. Changes in dynamic resting state network connectivity following aphasia therapy. Brain imaging and behavior, 12(4), pp.1141-1149.

Bonkhoff, A.K., Espinoza, F.A., Gazula, H., Vergara, V.M., Hensel, L., Michely, J., Paul, T., Rehme, A.K., Volz, L.J., Fink, G.R. and Calhoun, V.D., 2020. Acute ischaemic stroke alters the brain's preference for distinct dynamic connectivity states. *Brain*, 143(5), pp.1525-1540.

Bonkhoff, A.K., Schirmer, M.D., Bretzner, M., Etherton, M., Donahue, K., Tuozzo, C., Nardin, M., Giese, A.K., Wu, O., D. Calhoun, V. and Grefkes, C., 2021. Abnormal dynamic functional connectivity is linked to recovery after acute ischemic stroke. *Human brain mapping*, 42(7), pp.2278-2291.

Wang, Y., Wang, C., Miao, P., Liu, J., Wei, Y., Wu, L., Wang, K. and Cheng, J., 2020. An imbalance between functional segregation and integration in patients with pontine stroke: A dynamic functional network connectivity study. *NeuroImage: Clinical*, 28, p.102507.

- We thank the reviewer for pointing out these relevant references. While one of these articles (Bonkhoff 2021) was already featured in the original manuscript, all other references were missing. In the revised manuscript, we cite all these works in the discussion section, also providing a summary of their main findings that better allows contextualizing our work and showing where we improve over previous literature.

Methods/Results:

2. My main concern relates to the censoring preprocessing step. In general, 300 TRs, as utilized here, have certainly proven to be sufficient for dFNC analyses. However, given that each subject had six to eight scans with 128 TRs and therefore 768 or 1,024 TRs, those 300 TRs represent less than 50%? I am assuming all scans were shortened to 300TRs or did subjects have varying scan lengths? In what way were censored frames handled? Were there varying numbers of frames per window or some windows even empty in subjects? In what way were the multiple scans sessions integrated?

Head motion control has been shown to be very essential for dFNC analyses, but as Lurie et al (2020) point out, while there are sporadic positive reports (Laumann et al, 2017), censoring can interfere with robust dFNC estimation (Hutchinson et al, 2013; Zalesky et al, 2014). It is of prime importance for dFNC analyses to retain the correct temporal relationship between frames. Altogether, I currently find it difficult to evaluate the validity of any of the downstream dFNC analyses (e.g., do any of the observed group differences between healthy controls and patients arise due to varying censoring patterns?). The authors might consider conducting sensitivity analyses without censoring and other strategies for motion control, e.g., the inclusion of motion parameters as covariates during dFNC estimation.

- We agree with the reviewer that frame censoring has a potentially large impact on our results. In our work, we started from time series ts_0 with N time frames (with $N=6 \times 128$ or $N=7 \times 128$ or $N=8 \times 128$). We removed all censored time frames (let their number be M), obtaining time series ts_1 with $N-M$ time frames. Note that at this point, successive frames were in general no longer temporally adjacent (as some frames were temporally separated by removed frames). We then kept the first 300 points in each time series, obtaining time series ts_2 with 300 frames. This low number is the same for each subject. Finally, we applied sliding window analysis with fixed window length. This procedure amounts to keeping the number of frames per window constant, while varying the effective time span of each window. Some of the windows can include frames from different sessions, since we did not perform the analysis separately for different sessions.

To analyze the impact of frame censoring on our results, we performed control analyses to check for the impact of censoring (fig. R4). We recomputed dynamic measures (fraction times, dwell times and transition probabilities) with different censoring thresholds, a much more stringent threshold of 0.25, as well as a more liberal threshold of 0.75 (to facilitate comparison, dynamic measures were computed on the same set of DFS previously identified with a censoring threshold of 0.5). Results are shown in Fig. R4. Qualitatively, we did not observe important differences when varying the threshold. Quantitatively, among the three types of dynamical measures, transition probabilities were more strongly affected by the change in the threshold. However, fraction times were very robust. In particular, the relevant group differences identified in the main text do not depend on the threshold chosen.

3. Clustering: Taken from the results section: “To avoid one group dominating the other in the following clustering step, we equalized the number of controls and patients (since patients at different time points are considered different groups).” How many subjects were thus considered for the clustering step? All 20 + 20 controls and then how many patients per time point? Is it that you excluded patients simply for the clustering step and then later on assigned the excluded patients’ windows to specific DFSs that were derived from the harmonized group?

- We now better clarify this point in the Methods section at p. 26:

“We thus considered 20 controls at the first run, 20 controls at the second run and 47 patients at three time points – in total, $20+20+47\times 3=181$ sessions. To avoid biases, each session was shortened to 300 TR by excluding high-motion frames and keeping the first 300 good frames.”

4. Dynamic Functional States dynamics: Consider pointing out that “frequency of occurrence” and “average lifespan” correspond to the more commonly used terms “fraction time” and “dwell time” to increase the ease with which your results can be compared to other work.

- We adopted the nomenclature suggested by the reviewer.

It would furthermore be helpful, if you added a description of the split into “mild and severe FC impairment” groups, and how this FC impairment was defined, to the methods section. The split based on the FC impairment criterion, as described in the supplement, does not occur to be very straightforward to me, but may be fine from a methods perspective. With respect to any clinical relevance of findings, it might however be interesting to state the behavioral outcome averages per FC defined group. Are the patients with severe FC impairment also the ones that have the highest symptom load post-stroke?

- The split based on static FC has a strong correspondence with a clinical split based on the severity of the neurological symptoms. This is shown in SI paragraph S8, where we show that patients with $SPC>0$ were overall more severe in their neurological impairment than patients with $SPC<0$ (mean NIHSS score: 7.23 ($SPC>0$), 2.37 ($SPC<0$), T test: $T=4.02$, $p=2.610\cdot 10^{-4}$). These findings are consistent with previous work showing that subcortical stroke lesions cause at the same time stronger functional disconnection and more severe behavioral impairment than cortical lesions.

As a side note: To reduce ambiguity, It might be good to go through the entire manuscript and ensure that the groups are always referred to as high low “FC impairment” and not simply high/low “impairment”, as it might be confused with actual behavioral impairment.

- For the sake of simplicity, and due to the strong correspondence with clinical severity discussed above, we decided to simply refer to these two groups as severe/mild stroke patients.

Could you also describe more in detail which tests were used to examine which group differences? In what way did you incorporate mathematically that these were measurements from the same subjects over time?

- Whenever we jointly considered patients at different clinical stages within the same analysis, we used ANOVAs with repeated measures, which explicitly accounts for the fact that measures refer to the same subjects.

Did you check whether your input variables to the GLME model (a 2-way ANOVA?) were correlated? In my experience, group differences in derived dFNC measures, like fraction and dwell times are more commonly tested in 1-way ANOVAs, i.e., separately for each state and not additionally across states in a 2-way ANOVA, as fraction and dwell times of different states are usually highly correlated (c.f., Fu et al. 2019).

- We did not check that input variables were uncorrelated. However, the most relevant finding from this analysis are the post-hoc tests that identified group differences, and were performed separately for each DFS.

5. Cortical vs. subcortical pattern reorganization: I find this analysis particularly intriguing. However, given the novelty of this approach, I would suggest adding more details describing the methodological steps in the actual methods/results sections.

It would for example be good to mention the thresholding step that leads to the collection of subcortical regions in the main text (as it’s only mentioned in Figure 3 so far?)

Otherwise, it appears confusing why the subcortical regions are combined to “clusters” (as PCA-weights are continuous). How was the threshold of 0.2 chosen?

- We added some more detail in the main text to better highlight this point:

“We performed a Principal Component Analysis (PCA) of the leading eigenvector time-courses of subcortical regions (Fig.3C). The first two components explained 27% and 16% of the variance respectively. Since the two components projected onto different sets of regions, we defined two subcortical clusters: regions with loading greater than a threshold (0.2) on the first or second PC were assigned to cluster 1 or 2, respectively.”

The threshold of 0.2 was chosen upon inspection of the PCA results.

Is the brainstem assigned to both subcortical clusters?

- Yes, the brainstem was considered part of both clusters since both PCs loaded on the brain stem. In this sense, “clustering” based on the PC loadings can be considered as a fuzzy clustering which ends up with a single cluster assignment for most (but not necessarily all) regions.

Additionally: It would seem more straightforward to me, if the organization of cortical and subcortical regions into networks/clusters was performed similarly. At the moment, cortical regions are organized into a priori defined networks, while subcortical networks are estimated from the dynamic connectivity data at hand. The authors could consider discussing what effect this methodological difference might have had, e.g. in the Limitations section.

- We agree with this remark. Unfortunately, the strong clustering of subcortical regions was a result of analysis, and we had no a priori expectation that subcortical regions would broadly split into two main clusters. With hindsight, it would have been more straightforward to identify two subcortical networks based on static FC (which would yield essentially the same split into basal ganglia vs limbic regions).

*Shifts in cortical connectivity per network are evaluated based on changes between connectivity *averages* for each network and window. Is this the same for subcortical clusters, i.e. the connectivity of all within-cluster connections was averaged? Or was some PCA-weight used?*

- We now detail the procedure in the Methods section:

“For each DFS, we computed a measure of connectivity between subcortical and cortical networks (see **Fig. 3D**). To this aim, we considered the sub-matrix $v_{net} \times v_{sub}^T$ of $v_i \times v_i^T$, and we projected it onto the principal component space taking $v_{net} \times v_{sub}^T P^T$. In addition, we defined network-wise shifts in connectivity by computing the absolute value of the difference of v_{net} (for cortical networks) or Pv_{sub} (for subcortical principal components) between two consecutive sliding windows.”

How was the similarity between cortical and subcortical networks/clusters determined? Given Figure 3D, it's likely the average connectivity between cortical network and subcortical cluster?

- We now clarified this in the caption of Fig. 4:

“The average connectivity was computed as the Pearson correlation between the time course of the average network-wise connectivity (i.e., the average of of the (principal) submatrix of $v_i \times v_i^T$ corresponding to each network) and the time course of each subcortical PC similarity of time courses of average V in net vs subcortical PC”

Does Figure 4 exemplarily display the time courses of two individual subjects? How were these excerpts chosen?

- Figure 4 represents the time courses of two random subjects (analogous results are obtained for different subjects, i.e., jumps during state switches are evident in all subjects). We clarified this point in the caption of Figure 4.

Were there any group differences in these shift patterns between stroke patients and healthy controls?

- We ran a One-Way Anova to compare the number of subcortical shifts in the three groups (controls, severe acute patients and mild acute patients) and we did not find any significance ($F=2.4$, $p=0.10$).

6. *Relationship with anatomical lesions and structural disconnections: Great idea! In view of the double-multivariate nature of the structural vs. functional data, it might be an optimal use case for a method that is optimized for such a scenario, such as CCA or partial least square regression (c.f., Wang et al, 2020)?*

- The main reason why we did not adopt a double-multivariate approach is that we wanted to take advantage of the DFSs analysis. Once DFSs are obtained, FC dynamics is effectively reduced to the alternation between different states. Such alternation can be characterized at the individual level by summary statistics such as fraction times, dwell times and transition probabilities. All the multivariate information is encoded in the profile of the states – which, however, is fixed across subjects. Therefore, it is not immediate to combine DFS analysis with a double-multivariate approach.

In case the current approach is kept, I would recommend to perform the hyperparameter optimization in a nested cross-validation loop, as the performance estimates can be too optimistic otherwise (c.f., Varoquaux et al, 2017).

- Consistently with our previous papers (Salvalaggio, Alessandro, et al. "Post-stroke deficit prediction from lesion and indirect structural and functional disconnection." *Brain* 143.7 (2020): 2173-2188; Pini, Lorenzo, et al. "A novel stroke lesion network mapping approach: improved accuracy yet still low deficit prediction." *Brain communications* 3.4 (2021): fcab259.) we did not use a nested validation loop. The main rationale for using a nested cross-validation loop would be to ensure generalization of results to a different data set, and it would be necessary to *predict* dynamical PCs from lesions in new subjects. However, our main goal here is to establish whether lesions can *explain* dynamical PCs within the current data set (without aiming to generalize results to other data sets).

Additionally, consider describing how the dynamic PCAs were derived in the main text (currently mostly found in Figure 1?).

- This point is described in detail in the SI paragraph S9. We better clarified this point in the main text:

“We performed a PCA on fraction times, dwell times and transition probabilities - 35 dynamical features per subject. Three dynamic PCs (Dyn-PC) explained about 43% of the variability across subjects.”

Were the beta-weights corrected for multiple comparisons?

- We did not perform multiple comparison correction for beta weights. However, the classification of some weights as “significant” was mainly done for visualization purposes.

7. *Relationship with behavior: I would suggest describing more in detail in what way FC (and dFNC) information was integrated in these analyses. Did you only use the first two FC PCA-components that were also used to define the split in two $SPC > 0$ and < 0 groups?*

- As explained in detail in the Methods section, we used as regressors the first PC obtained in the principal component analysis of static FC - which was also used to split patients into “mild” and “severe”- and the first three PCs obtained in the principal components analysis of the dFC measures (fraction times, dwell times, transition probabilities) – which was used in the analysis of the relation between lesions and dFC.

If so, it would be good to render it more apparent that these analyses compare two very specific aspects of FC and dFNC data (and findings might or might not hold for other representations of FC and dFNC information).

- We added a paragraph in the Limitations subsection of the Discussion section to make this point:

“Finally, to study the relation between FC/dFC and lesions/behavior we made a radical dimensionality reduction step: multivariate information about FC/dFC was effectively condensed in a short array of scalar quantities: three dynamical principal components, summarizing the fraction times, dwell times and transition probabilities of different DFSs; and one static principal component, summarizing the most common pattern of anomalous FC in stroke patients. These scalar quantities cannot be assumed to faithfully represent all possibly relevant aspects of FC and dFC. Therefore, our analysis may have missed significant relations between lesions/behavior and FC/dFC that may become apparent only when using different, more complex representations of FC and dFC information.”

Additionally, consider changing your language to “explain” (instead of “predict”) throughout the entire manuscript, as no prediction performance per se was tested (analyses were performed in-sample, weren’t they?)

- We agree with the remark, and we changed the wording in the manuscript.

8. Two previous dFNC in stroke studies (Bonkhoff et al., 2020, 2021) actually found an increase in network segregation in case of severe stroke and an increase in network integration in case of only moderate motor impairment. These findings of an increased network segregation are in line with the ones obtained from healthy subjects after cast-induced motor inactivity of the upper limb (Newbold et al., 2020). The authors could consider discussing these previous findings with respect to their own ones (previous studies have for example defined groups of stroke patients based on clinical symptoms post-stroke and not static functional connectivity).

- In our view, our findings are not in contradiction with those of Bonkhoff 2020, 2021. Indeed, we find that patients overexpress specific states, like DFS2, DFS4, presenting a *globally weaker* network segregation. Note, however, that this globally weak segregation is mostly concerning cognitive networks, and the same states feature strong segregation of specific networks: most notably, in DFS2 the visual network is strongly segregated from the rest of the brain (actually, this feature is also present in state 1 in Bonkhoff 2021, and it is even more prominent there since the relative number of visual regions is much larger than in our study; in fact, DFS2 in our study and state1 in Bonkhoff 2021 bear several resemblances). Interestingly, DFS2 is also the state with the highest segregation of the sensorimotor network, so that our results are also in accordance with those of Bonkhoff 2020, which points at a stronger segregation of the motor network after stroke.

We now briefly discuss this point in the discussion section

“Bonkhoff et al. identified three DFSs, one with strong segregation of VIS and SMN from other networks (akin to DFS2 in this work), one with weak correlations (akin to DFS4), and one with anticorrelations between visual-sensorimotor networks and DMN (akin to DFS1). Stroke patients overexpressed the first state: although Bonkhoff et al. characterized this state as a state of “anomalous segregation”, in fact segregation mostly occurs for the visual/SMN network, while cognitive networks are integrated (as in our DFS2).”

9. Limitation section: Consider discussing differences in the choice of exact methodology (many other dFNC papers would more commonly employ tapered windows, estimate dynamic connectivity via the precision matrix (Preti et al., 2017), use the l1-norm for k-means clustering (Aggarwal et al., 2001) and also perform more preprocessing steps like despiking). Would we expect any differences if analyses were repeated varying these aspects?)

- In our opinion, it is essentially impossible to fully explore the space of all methodological options available. Surely one may have performed the sliding windows + K-means analysis in various, slightly different ways, such as those listed by the Reviewer. However, one may have tried to characterize DFS with altogether different methodologies, such as, e.g., Hidden Markov Models, switching linear dynamical systems, etc. The main question is, what kind of consistency can we expect across methodologies? May we expect some results to be robust to the specific methodological choices made? In our opinion, some major qualitative findings may replicate in quite different methodological scenarios.

For instance, severe stroke patients overexpress dynamical states with low inter-hemispheric integration. We expect this finding to emerge irrespectively of the detailed DFS landscape, which depends on methodological choices. An instance of this can be observed if, keeping all other methodological choices fixed, we use a different number K of DFS ($K=10$ instead of $K=5$). In this case, the DFS landscape changes; however, states still differ in terms of the same FC features (homotopic FC, DAN-DMN FC, modularity). By comparing results with $K=5$ and $K=10$, we find that patients over-express the state(s) with low homotopic FC (which differ in the two cases). Another robust feature we may expect is that the two groups of subcortical regions identified will flexibly synchronize with different sets of cortical networks across different DFSs – regardless of the specific DFSs used.

In this sense, DFSs may be regarded as a suitable “state basis” (like a function basis) to analyze group differences in dynamic FC: different “bases” are possible, but results may easily map from one basis to another.

10. Please increase the font size to ensure all captions are legible.

- We increased the font size whenever appropriate to ensure legibility.

11. What does dynamic connectivity tell us that static connectivity did not?

- There are many observations that are a unique result of the dynamic analysis:

- a) That cortical and subcortical connectivity shifts are synchronized; this speaks to a large-scale coordination of cortical and subcortical activity that does not surface at all in static FC analysis.
- b) That static FC anomalies in patients mirror the “over-expression” vs “under-expression” of specific transient connectivity patterns
- c) That mild vs severe patients present qualitative differences in their dynamics
- d) That subcortical lesions are most important to explain dynamic anomalies

12. *It would be good to change “acute” to “subacute” throughout the entire manuscript (c.f., Bernhardt et al., 2017).*

- We changed wording as suggested by the Reviewer.

REVIEWERS' COMMENTS

Reviewer #1 (Remarks to the Author):

The authors have adequately addressed my concerns.

Reviewer #2 (Remarks to the Author):

I thank the authors for their revision in which they have addressed many of my previous concerns. I have several remaining remarks. Most of these are follow-up comments to my previous points. Given that Nature communications is such a widely read journal and oftentimes inspires future studies that will use similar methodological approaches, I would suggest to include more comprehensive changes to the actual manuscript and not only the revision letter. In my opinion, many important methodological aspects might be missed by readers otherwise. However, I am happy for the editors to decide whether and in what way these further changes are necessary.

- Censoring: It is indeed reassuring that fraction and dwell times, as well as transition probabilities remain qualitatively similar with different scan lengths. Please specify which relevant group differences remained the same. Furthermore, I think it would be equally important to repeat these sensitivity analyses for your main finding of cortical vs. subcortical reorganization pattern. The stability of results for fraction and dwell times etc. can't easily be generalized to this analysis and it would strengthen the credibility of your findings immensely. Further, censoring 25% of frames is still a large amount of data, which should be briefly discussed in the limitation section.

- Please specify which GLME model you employed to evaluate group differences in dynamical measures (page 26). In case it was indeed a 2-way ANOVA that took into account the fraction or dwell times for all states at the same time, please add to the manuscript that you didn't check for collinearity of input variables and that it is therefore difficult to determine whether the model was an appropriate starting point. As pointed out previously, other work usually employs 1-way ANOVAs that tests group differences for all states separately, as fraction and dwell times are expected to correlate quite strongly (c.f., e.g. Fu et al., Human Brain Mapping 2019).

- I apologize, but I admittedly don't understand the reply to my previous comment 6. Fraction and dwell times are available on single-subject level and could thus be fed in a double-multivariate model, couldn't they? Some of the authors did actually showcase such an approach in a recent publication in Neuroimage (Pirondini et al., 2022)? In any case, maybe mention in the limitation section, that CCA or partial least square correlation could have been valid alternatives.

- The authors should mention in their limitation section that they did not test for generalization of ridge regression findings to new samples and that there is a risk for current findings of ridge regression analyses to be too optimistic as the hyperparameter optimization was not performed with nested cross-validation.

- Given that the authors confirmed that no prediction analyses were performed and that "predict" should be changed to "explain", I would recommend to proofread the manuscript once again to ensure this change is consistently implemented (especially in the abstract).

Reply to Reviewer #1

We thank Reviewer #1 for her/his appreciation of our work.

Reply to Reviewer #2

We thank Reviewer #2 for her/his appreciation of our work and her/his carefulness in asking for methodological rigor and clarity.

In our revised manuscript, we have addressed her/his remaining concerns.

Censoring: It is indeed reassuring that fraction and dwell times, as well as transition probabilities remain qualitatively similar with different scan lengths. Please specify which relevant group differences remained the same. Furthermore, I think it would be equally important to repeat these sensitivity analyses for your main finding of cortical vs. subcortical reorganization pattern. The stability of results for fraction and dwell times etc. can't easily be generalized to this analysis and it would strengthen the credibility of your findings immensely. Further, censoring 25% of frames is still a large amount of data, which should be briefly discussed in the limitation section.

We added a paragraph (SI-Paragraph S5) where we mention the robustness of group differences:

“To be more precise, with threshold = 0.25 all differences in fraction times between CTRs and severe sub-acute patients (DFS1, DFS2, DFS3, and DFS4), as well as the difference in DFS5 between CTR and mild sub-acute patients still hold, after FDR correction. With threshold = 0.75, we confirmed the significant difference in fraction times for DFS2 and DFS3 between CTRs and severe sub-acute patients.”

We included an additional control analysis (SI-Paragraph S5, SI-Fig. 9, SI-Fig. 10) showing the robustness of the cortical vs. subcortical reorganization pattern with respect to censoring.

Finally, in the Discussion section, we mention the potentially large impact of motion scrubbing:

“Finally, due to the long TR and the large impact of motion scrubbing (on average, 25% of data points are discarded), the amount of data available per subject is limited, which limits the reliability of individual estimates of dynamic FC metrics. Therefore, while our group results indicate that dynamic FC metrics are correlated with stroke severity, their use as individual biomarkers is currently limited.”

Please specify which GLME model you employed to evaluate group differences in dynamical measures (page 26). In case it was indeed a 2-way ANOVA that took into account the fraction or dwell times for all states at the same time, please add to the manuscript that you didn't check for collinearity of input variables and that it is therefore difficult to determine whether the model was an appropriate starting point. As pointed out previously, other work usually employs 1-way ANOVAs that tests group differences for all states separately, as fraction and dwell times are expected to correlate quite strongly (c.f., e.g. Fu et al., Human Brain Mapping 2019).

We clarify that we were indeed using a factorial design, and we mention the issue raised by the Reviewer in (l. 249):

“We tested the effect of DFSs and groups (controls, sub-acute severe patients and sub-acute mild patients) in the fraction times through a Generalized Linear Mixed Effect Model (GLME) with Poisson distribution, with DFS and group as factors. We found a significant interaction effect ($F = 211.13, p = 0$) and both single main effects of DFSs ($F = 151.95, p = 0$) and groups ($F = 168.29, p = 0$). Note that this result may be affected by collinearity, as fraction times for different DFS are not independent. However, we conducted post-hoc analyses for each DFS separately (with non-parametric permutation tests), finding two different patterns of abnormalities for severe and mild patients.”

I apologize, but I admittedly don't understand the reply to my previous comment 6. Fraction and dwell times are available on single-subject level and could thus be fed in a double-multivariate model, couldn't they? Some of the authors did actually showcase such an approach in a recent publication in Neuroimage (Pirondini et al., 2022)? In any case, maybe mention in the limitation section, that CCA or partial least square correlation could have been valid alternatives.

In our previous Reply, we misunderstood the Reviewer's comment. In fact, we thought that the Reviewer was suggesting to related the full, high-dimensional structural data with the full, high-dimensional functional data (time-dependent FC) matrices. In fact, the Reviewer correctly points out that also the array of fraction or dwell times represents multivariate data, and would be suitable to be included in a double multivariate analysis together with CCA or PLS. We mentioned this point in the Discussion section (l. 503)

“In principle, a possible alternative to this large dimensionality reduction would have been to use double-multivariate methods such as partial least squares or canonical correlation analysis, but we are unsure whether this would have led to easily interpretable results.”

The authors should mention in their limitation section that they did not test for generalization of ridge regression findings to new samples and that there is a risk for current findings of ridge regression analyses to be too optimistic as the hyperparameter optimization was not performed with nested cross-validation.

We mention these limitations in the Discussion section (l. 506):

“Moreover, we did not test for generalization of ridge regression findings to new samples, therefore current findings of ridge regression analyses may be specific to the used subjects' sample. While performing nested cross-validation may enhance robustness of these findings, we believe that conclusive evidence may be obtained only by replicating these findings in an independent subject cohort.”

Given that the authors confirmed that no prediction analyses were performed and that “predict” should be changed to “explain”, I would recommend to proofread the manuscript once again to ensure this change is consistently implemented (especially in the abstract).

We checked the manuscript, and removed the working “prediction” as well as any prediction claim.